# AbDiffuser: Full-Atom Generation of in vitro Functioning Antibodies

**Karolis Martinkus**[1], **Jan Ludwiczak**[1], **Kyunghyun Cho**[1,4], **Wei-Ching Liang**[2],
**Julien Lafrance-Vanasse**[2], **Isidro Hotzel**[2], **Arvind Rajpal**[2], **Yan Wu**[2], **Richard Bonneau**[1],
**Vladimir Gligorijevic**[1], **Andreas Loukas**[1]

[1]Prescient Design, Genentech, Roche [2]Antibody Engineering, Genentech, Roche [4]NYU

## Abstract

We introduce *AbDiffuser*, an equivariant and physics-informed diffusion model for the joint generation of antibody 3D structures and sequences. AbDiffuser is built on top of a new representation of protein structure, relies on a novel architecture for aligned proteins, and utilizes strong diffusion priors to improve the denoising process. Our approach improves protein diffusion by taking advantage of domain knowledge and physics-based constraints; handles sequence-length changes; and reduces memory complexity by an order of magnitude, enabling backbone and side chain generation. We validate AbDiffuser *in silico* and *in vitro*. Numerical experiments showcase the ability of AbDiffuser to generate antibodies that closely track the sequence and structural properties of a reference set. Laboratory experiments confirm that all 16 HER2 antibodies discovered were expressed at high levels and that 57.1% of the selected designs were tight binders.

## 1 Introduction

We focus on the generation of immunoglobulin proteins, also known as antibodies, that help the immune system recognize and neutralize pathogens. Due to their potency and versatility, antibodies constitute one of the most popular drug modalities, with 10 out of 37 newly FDA-approved drugs in 2022 being immunoglobulins [58]. The ability to generate new antibodies with pre-defined biochemical properties *in silico* carries the promise of speeding up the drug design process.

Several works have attempted to generate antibodies by learning to form new sequences that resemble those found in nature [21; 19; 82]. An issue with sequence-based approaches is that it is hard to determine the properties that render a protein a functional molecule (and an antibody a potent drug) without inspecting a 3D model of the functional state such as an interface or active site. So far, almost all of the first design methods that have enabled novel protein design used carefully curated structure-function information to score the designs [45; 48; 30]. The determinant role of structure on function has motivated numerous works to co-design sequence and structure [4; 24; 89; 34; 43; 3; 51] or to first design the protein backbone and then fill in the amino acid identities [90; 32; 15; 87; 50; 92; 47].

An emerging paradigm for protein generation is that of equivariant diffusion [90; 89; 51]. Protein diffusion models combine ideas from equivariant and physics-informed protein structure representations [37] with advances in denoising diffusion probabilistic models [28] to gradually transform noise to a partial structure. The generated structure is then refined by adding side chains and optimizing atom positions to form a full-atom 3D model of a protein.

A pertinent challenge when generating protein structure is satisfying the appropriate equivariance and physics constraints while also balancing modeling complexity and fidelity. Most current protein models rely on equivariant transformers [37; 73] or graph neural networks [11; 43] to satisfy SE(3)

---

Correspondence to Andreas Loukas <`andreas.loukas@roche.com`>.

37th Conference on Neural Information Processing Systems (NeurIPS 2023).

equivariance and typically represent parts of the protein geometry in angle space. The latter recipe can be set up to build a physics-informed model that respects (many) desired constraints but comes with increased complexity. As a consequence, these models are expensive to train and are often applied to a partial protein, ignoring the side chain placement [43; 51] or even deferring the determination of amino acid identity to a later stage [90; 32]. Alternatively, some works focus on the regeneration of complementarity-determining regions (CDRs) of antibodies that are of particular importance to binding [34; 43; 33; 53; 78; 44], which also helps to reduce the complexity of the problem.

Our work is motivated by the observation that key large protein families, here we focus on the antibody protein family, typically have strong properties, such as an ability to be mapped to a reliable sequence ordinate via sequence alignment. Our main contribution is an equivariant diffusion model called *AbDiffuser* that is designed to exploit these properties. We show that incorporating family-specific priors into the diffusion process significantly improves generation efficiency and quality.

AbDiffuser relies on a new *universal* SE(3) equivariant neural network that we call the Aligned Protein Mixer (APMixer). In contrast to existing equivariant architectures used for antibodies and proteins, APMixer models residue-to-residue relations implicitly and is particularly effective in handling sequence length changes. Additionally, its significantly smaller memory footprint makes it possible to generate full antibody structures, including framework, CDRs, and side chains. Our approach to residue representation is made possible by a projection method guaranteeing that bond and angle constraints are respected while operating in coordinate and not angle space, a better match to diffusion with Gaussian noise. We also benefit from the effects of overparameterization by scaling the network size to hundreds of millions of parameters on a single GPU; an order of magnitude improvement over the corresponding (E)GNN architectures. Having a powerful model that generates the full antibody structure is shown to be beneficial for the quality of the designed proteins.

We evaluate AbDiffuser on the generation of antibodies from paired Observable Antibody Space (pOAS) [61], modeling of HER2 binders and antigen-conditioned CDR redesign. Numerical experiments demonstrate that the proposed representation, architecture, and priors enable AbDiffuser to better model the sequence and structural properties of natural antibodies. We also submitted a subset of our proposed HER2 binders for experimental validation in a laboratory. Of the 16 samples submitted, 100% were expressed as actual antibodies and 37. 5% bound to the target with an average pKD of 8.7. Of these, the subset containing raw/filtered samples achieved a binding rate of 22.2%/57.1% and an average pKD of 8.53/8.78, with our tightest binder slightly improving upon the affinity of the cancer drug Trastuzumab. These results provide the first experimental evidence that a generative model trained on mutagenesis data can reliably (with high probability) create new antibody binders of high affinity, even without post-selection by learned or physics-based binding predictors.

Due to space limitations, we refer the reader to the appendix for a detailed discussion of related work, for method details including proofs, as well as for additional numerical results.

## 2 Denoising Diffusion for Protein Generation

This section describes how we utilize denoising diffusion to generate antibody sequence and structure. The ideas presented are influenced by previous work on denoising diffusion [26; 5; 28]. The main contributions that distinguish our work are presented in Sections 3 and 4.

We adopt the denoising diffusion framework in which, given a data point $X_0$, the forward diffusion process gradually adds noise to form corrupted samples $X_t$. These samples form a trajectory $(X_0, X_1, ..., X_t, ..., X_T)$ of increasingly noisy data, interpolating from the data distribution $X_0 \sim p(X_0)$ to that of an easy-to-sample prior $X_T \sim p(X_T)$, such as a Gaussian. The process is constructed to be Markovian, so that $q(X_2, ..., X_T|X_0) = q(X_1|X_0) \Pi_{t=2}^{T} q(X_t|X_{t-1})$.

To generate new data, a neural network learns to approximate the true denoising process $\hat{X}_0 = \phi(X_t, t)$, which can be achieved by minimizing the variational upper bound of the negative log-likelihood [84; 26; 5]. In our diffusion process, we factorize the posterior probability distribution over the atom positions and residue types:

$$q(X_t|X_{t-1}) = q(X_t^{\text{pos}}|X_{t-1}^{\text{pos}}) \, q(X_t^{\text{res}}|X_{t-1}^{\text{res}}),$$

where a Gaussian and a categorical distribution govern the noise applied to atom positions and residue types, respectively. We cover these in detail in Sections B.2 and B.3. The reverse diffusion process is

joint: the model jointly considers atom positions and residue types $(\hat{X}_0^{\text{pos}}, \hat{X}_0^{\text{res}}) = \phi(X_t^{\text{pos}}, X_t^{\text{res}}, t)$. Throughout the paper, we use $X^{\text{pos}} \in \mathbb{R}^{n \times 3}$ as a matrix of antibody atom positions and $X^{\text{res}} \in \mathbb{R}^{r \times 21}$ as a matrix of one-hot encodings of residue types (20 amino acids and a gap).

# 3 Aligned Protein Mixer

We present Aligned Protein Mixer (APMixer), a novel neural network for processing proteins from a family of aligned proteins. In particular, we focus on antibodies, although the method can in principle be applied to any sufficiently large protein family by using a family-specific global alignment [81; 57]. We first explain how we ensure SE(3) equivariance and how we account for variable sequence lengths in Sections 3.1 and 3.2. The model is detailed in Section 3.3 and our approach to incorporate physics-based constraints on bond angles and lengths is described in Section 3.4.

## 3.1 SE(3) Equivariance by Frame Averaging

Any neural network $\phi$ whose input $X$ has dimension $n \times 3$ can be made equivariant or invariant to group transformations by averaging the model outputs over a carefully selected subset of group elements called frames $\mathcal{F}(X)$ [68]. For example, this has been used successfully to make equivariant self-attention models for the prediction of protein-ligand binding energy [35].

We achieve the desired equivariance to rotations and translations (the SE(3) group) as follows:

$$X^{\text{pos}} = \frac{1}{|\mathcal{F}(X^{\text{pos}})|} \sum_{(R,t) \in \mathcal{F}(X^{\text{pos}})} \phi(X^{\text{pos}} R - \mathbf{1}t, X^{\text{res}}) R^T + \mathbf{1}t, \tag{1}$$

where $t = \frac{1}{n} \mathbf{1}^T X^{\text{pos}}$ is the centroid of our points and the four canonical rotation matrices $R$ forming the four frames $\mathcal{F}(X^{\text{pos}}) \subset \text{SE}(3)$ needed to achieve equivariance can be determined based on Principle Component Analysis. More specifically, we obtain three unit-length eigenvectors $v_1, v_2, v_3$ corresponding to the eigenvalues $\lambda_1, \lambda_2, \lambda_3$ from the eigendecomposition of the covariance matrix $C = (X^{\text{pos}} - \mathbf{1}t)^T (X^{\text{pos}} - \mathbf{1}t) \in \mathbb{R}^{3 \times 3}$ and define the four frames as

$$\mathcal{F}(X^{\text{pos}}) = \left\{ ([\alpha v_1, \beta v_2, \alpha v_1 \times \beta v_2], t) \mid \alpha, \beta \in \{-1, 1\} \right\}.$$

To respect the fixed chirality of proteins observed in humans, we desire equivariance w.r.t. SE(3) and not E(3) which also includes reflections. As such, when constructing frames the third axis sign is not varied but its direction is determined by the right-hand rule (cross product of the first two axes).

SE(3) invariance can be similarly achieved:

$$X^{\text{res}} = \frac{1}{|\mathcal{F}(X^{\text{pos}})|} \sum_{(R,t) \in \mathcal{F}(X^{\text{pos}})} \phi(X^{\text{pos}} R - \mathbf{1}t, X^{\text{res}}) \tag{2}$$

We make use of equation 2 when denoising residue types $X^{\text{res}}$ as they are invariant to the rotation of the antibody, and equation 1 for the prediction of atom positions $X^{\text{pos}}$.

## 3.2 Handling Length Changes by Multiple Sequence Alignment

The version of APMixer investigated in this work is built on top of the AHo antibody residue numbering scheme proposed by Honegger and Plückthun [27]. This numbering scheme was constructed in a data-driven fashion using information from known antibody structures. For each residue in an antibody chain, it assigns an integer position in $[1, 149]$ based on the structural role of the residue (e.g. being in a particular CDR loop or in the framework region between two particular CDR loops). Essentially all known antibodies fit into this representation [27].

As we are modeling paired antibody sequences (the heavy and light chain), the full representation is a $2 \times 149$ element sequence, where 149 heavy chain elements are followed by 149 light chain elements. We represent AHo gaps physically as 'ghost' residue placeholders; their position is determined in data pre-processing by linearly interpolating between the corresponding atoms of the nearest existing residues (trivial, due to the use of AHo numbering).

Our experiments confirm that the proposed representation consistently improves generation quality. The reason is two-fold: a) Each antibody chain is now represented as a fixed-length sequence with

149 positions that are either empty gaps or are filled with an appropriate amino acid. This fixed-length representation encompasses antibodies with diverse loop lengths via alignment and thus allows our generative model to internally choose how many residues the generated protein will have. In contrast, many of the non-sequence-based protein, small molecule, and graph generative models [49; 54; 28; 90; 32] require the number of elements in the object to be specified beforehand. b) The sequence position directly implies the structural role that the amino acid needs to perform, which makes it easier for the model to pick up structure-specific rules.

### 3.3 The APMixer Architecture

Our use of a fixed length representation for the immunoglobin variable domain fold family (Section 3.2) also allows us to forgo traditional architecture choices in favor of a more efficient architecture inspired by the MLP-Mixer [86]. We build the model architecture out of blocks, where each block consists of two MLPs, that are applied consecutively on the columns and the rows

$$X_{\cdot,j} = X_{\cdot,j} + W_2\rho(W_1\text{LayerNorm}(X_{\cdot,j}))\text{for all } j \in [c]$$
$$X_{i,\cdot} = X_{i,\cdot} + W_4\rho(W_3\text{LayerNorm}(X_{i,\cdot}))\text{for all } i \in [r],$$

of the input matrix $X \in \mathbb{R}^{r \times c}$, with $\rho$ being an activation function, and using the notation $[k] = (1, \ldots, k)$. We define the model input as a fixed-size matrix $X$ combining $X^{\text{pos}}$ and $X^{\text{res}}$ with one sequence element per row ($2 \times 149$ rows in total). In each row, we encode the residue type and all of the atom positions for that residue (e.g., the $C, C_\alpha, N, C_\beta$ backbone atoms). These input matrix rows are embedded in higher-dimensional vectors using an MLP. Specifically, using this representation, our input matrix $X$ has $r = 2 \times 149$ rows and $c = 21 + 4 \times 3$ columns.

To achieve equivariance to atom translations and rotations, we can either apply frame averaging (Section 3.1) on the whole model or on each AbDiffuser block individually. We chose the second option as in our preliminary experiments this improved performance. Frame averaging can be applied on high-dimensional embeddings simply by splitting them into three-dimensional sub-vectors and using each to compute the SE(3) frames [68]. To account for that residue types are invariant to Euclidean transformations, while the atom positions are equivariant, we split each block's input and output vectors in half, with one half being treated equivariantly and the other invariantly.

**Model complexity and SE(3)-universality.** APMixer models pairwise interactions implicitly by operating on rows and columns of the input interchangeably. Thus, its memory complexity grows *linearly* with the number of residues. This contrasts with the usual quadratic complexity of traditional structure-based models and allows us to do more with a fixed run-time, parameter, and/or memory budget. In Appendix G we prove that the model on top of which APMixer is built is SE(3)-universal, meaning that it can approximate any SE(3)-equivariant function.

We also remark that, in principle, other models, such as a 1D CNN or a transformer, could be used in place of the MLPs in APMixer. With such sequence-length-independent models, we would no longer require a multiple sequence alignment of the given protein family, though this would possibly come at the expense of universality and efficiency.

### 3.4 Physics-informed Residue Representation by Projection

Atoms within a protein adhere to strong constraints. In principle, a neural network trained with enough data can learn to respect these constraints. However, over a fixed data budget, it can be advantageous to construct a model in a manner that guarantees that its outputs never violate the constraints. Previous work commonly represents proteins in terms of rotation and translation of rigid residue frames and uses idealized residue representations to recover atom positions [37; 73]. Although there is a long tradition of modeling backbone and side chain degrees of freedom in angle space [67; 14], operating in angle space adds modeling complexity and makes diffusion potentially inaccurate [94; 8; 90]. We take a different route and devise a way to respect bond constraints while operating in a global coordinate frame, which works seamlessly with standard Gaussian diffusion for atom positions.

Specifically, inspired by interior-point methods [72] that alternate between optimization and projection steps, we design a new non-parametric projection layer that is applied to both model inputs and outputs. The model and the noising process are allowed to move the atoms freely, and the projection layer then corrects their positions such that the constraints are respected.

**Backbone residue projection.** We use a reference residue backbone $(C, C_\alpha, N, C_\beta)$ with idealized bond lengths and angles [17]. As the backbone atoms are rigid, we rely on the Kabsch algorithm [38] to identify the optimal roto-translation between the projection layer's input and the rigid reference residue's atom positions. We apply the transformation to the reference residue and output the resulting atom positions. We also ensure that the distance between the corresponding $C$ and $O$ atoms in the output is $1.231$Åwhile staying as close as possible to the input $O$ position. The reference residue is also used to represent the AHo ghost residue atoms. Idealizing a backbone in this way usually results in a negligible RMSE to the original structure of $\sim 5 \cdot 10^{-3}$Å.

**Side chain projection.** We employ a similar idea to constrain the placement of side chains. In contrast to structure prediction [37; 46], we cannot use one idealized side chain per amino acid, since the sequence is unknown during generation. Our solution is to convert all side chains to a generic representation with enough degrees of freedom to account for the exact placement of all atoms. The amino acid side chains have up to 4 bonds that they can rotate around by dihedral angles. These degrees of freedom can be captured by constructing a side-chain template that consists of 4 pseudo-carbon atoms, for which the dihedral angles are rotated in the same way as for the original side chain. If the original side chain has fewer degrees of freedom, we simply set the corresponding dihedral angles to $180°$ such that the atoms forming the dihedral angle lie on a plane. The projection layer then only has to ensure that the bond lengths between the pseudo-atoms are respected. We set the bond length to $1.54$Å because carbon atoms are the most common atoms that form the dihedral angles of real amino acids. This representation can be seen in Figure 1. To recover the full-atom structure, we extract the dihedral angles from the side chain template; the angles are then applied to idealized amino acid-specific templates.

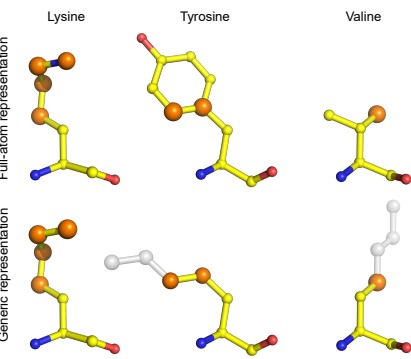

Figure 1: The proposed internal generic side chain representation. The dihedral-defining atoms (orange) from the full-atom representation (top) are used to construct a generic four-atom representation (bottom). If the side chain has fewer than four angles, additional atoms (gray) are placed in the generic side chain to correspond to a $180°$ angle. The full atom representation is recovered by applying matching rotations to an appropriate side chain template.

## 4 Informative Diffusion Priors

It has been hypothesized that diffusion processes for generation should be tailored to the particular task to increase performance [88; 5]. In Appendix E.1, we present a theoretical argument confirming that choosing a good prior reduces the complexity of the learnable model needed to achieve good-quality denoising. Our bound stated in Theorem E.1 reveals that one cannot learn a simple generative model that fits the data well unless the prior has a small Wasserstein distance to the data distribution.

Armed with this knowledge, we next introduce two types of priors that are incorporated into our diffusion model: a) position-specific residue frequencies that describe sequence conservation of the immunoglobulin fold, and b) conditional dependencies between atom positions.

### 4.1 AHo-specific Residue Frequencies

A separate benefit of having a fixed-length sequence representation (see Section 3.2) is that we can use the residue frequency at each position as a diffusion prior. To do this, we estimate marginal position-specific categorical distributions $Q^1, ..., Q^{2*149}$ over residue types from the training data and use these to define the noise in the discrete residue-type diffusion (see Appendix B.3). Since AHo aligns residues based on their structural role within the protein, the estimated prior distributions exhibit significant variability across positions and have low entropy in preserved regions of the immunoglobulin fold. In this way, the noise $q(X_t^{\text{res}}|X_0^{\text{res}}) = \text{Cat}(X_0^{\text{res}}(\beta_t I + (1 - \beta_t)Q^i))$ at every step of the forward process depends on the residue position $i$ in the fixed-length sequence representation. Gap frequencies are also captured in these noise matrices.

### 4.2 Encoding Conditional Atom Dependencies

It is a consequence of their chain-like structure that neighboring residue atom positions of proteins are strongly correlated. Theorem E.1 suggests that encoding these correlations within the diffusion process eases the denoising complexity and can free the model to focus on more challenging aspects.

To this end, we capture the conditional independence relations between atom positions by a Gaussian Markov Random Field (GMRF) [42]. The latter corresponds to a Gaussian $\mathcal{N}(0, \Sigma)$ over atom positions whose precision matrix $\Sigma^{-1} = L + aI$ is equal to the shifted graph Laplacian $L = \text{diag}(A\mathbf{1}) - A$ associated with the adjacency matrix $A$. The GMRF operates under the assumption that node features (e.g., 3D positions) for which there is no edge in the adjacency $A$ are conditionally independent. Some concurrent works [32; 36] also considered conditional atom dependencies by hand-crafting correlation matrices that capture chain relations between residues. We take a step further by proposing to automatically learn the sparse conditional dependence relations from the training data. Our approach entails estimating a sparse adjacency matrix $A$ that captures the data variance under the GMRF model. The details are described in Appendix F.

## 5 Experiments

We evaluate AbDiffuser's ability to design antibodies. After describing the experimental setup and metrics, Section 5.1 presents *in silico* tests illustrating the effect of our proposed changes on the generation quality. Section 5.2 details *in vitro* results showcasing the ability of our method to design new expressing antibodies that bind to a known antigen. Further analyses can be found in the Appendices K and L, whereas experiments on CDR redesign in SAbDab are in Appendix N.

**Baselines.** We compare APMixer with: a) a sequence transformer based on BERT [12; 74] whose output is folded to recover the structure; b) an E(n) Equivariant Graph Neural Network (EGNN) [75] which is a popular choice for tasks such as 3D molecule generation [28], antibody CDR loop inpainting [43], and antibody generation [89]; and c) a FA-GNN [68], corresponding to a standard GNN with SE(3) equivariance attained by frame averaging. We also evaluate the proposed informative antibody-specific priors using all of these architectures. To ensure the comparison is performed fairly and under similar settings, we always use the projection layer and our diffusion implementation only varying the denoising model architecture. In the OAS task, we also compare our diffusion-based approach with d) the IgLM [80] antibody language model. To ensure that it generates paired OAS-like sequences, we condition the generation on the subsequences of the first and last few amino acids taken from pOAS sequences (something that our approach does not need). It is also important to note that the comparison is not fair, since the publicly available IgLM model was trained on 558M sequences that also include the whole paired OAS (105k sequences) and the test set we use. So, in many ways, IgLM's performance represents the best results we could ever hope to achieve with a sequence-only approach. We also compare with e) dyMEAN [44], which is the only other antibody generative model previously shown to be able to jointly generate the full strucutre and sequence. In the binder generation task, we compare with f) RefineGNN [34], g) MEAN [43] and h) DiffAb [53], three state-of-the-art geometric deep learning methods for CDR redesign.

**Metrics.** The quality of generated sequences is measured in terms of their *naturalness* (inverse perplexity of the AntiBERTy [74] model), *closeness* to the closest antibody in the training set in terms of edit distance, and *stability* estimated by IgFold. We also verify that the generated antibody sequences satisfy the appropriate biophysical properties using four additional structure-based metrics [69]: CDR region hydrophobicity (*CDR PSH*), patches of positive (*CDR PPC*), and negative charge (*CDR PNC*), and symmetry of electrostatic charges of heavy and light chains (*SFV CSP*). The metrics applied to generated structures focus primarily on the estimated free energy $\Delta G$ using Rosetta [2] and *RMSD* for backbone heavy atom positions as compared to IgFold [73] predictions. More details can be found in Appendix J.

As we want to estimate how well the entire data distribution is captured, in all cases except RMSD, we report the Wasserstein distance between the scores of the sequences in the test split and the scores of the generated sequences. As a reference, we also report the baseline metrics achieved by the sequences and structures in the validation set. A generative model that approaches or matches these values is effectively as good at modeling the distribution of the specific metric in consideration as i.i.d. sampling. The test set and the generated set always have 1000 examples.

Further details on training and implementation can be found in Appendix M.

| Model | $W_1$(Nat.)↓ | $W_1$(Clo.)↓ | $W_1$(Sta.)↓ | $W_1$(PSH)↓ | $W_1$(PPC)↓ | $W_1$(PNC)↓ | $W_1$(CSP)↓ | $W_1(\Delta G)$↓ | RMSD↓ |
|---|---|---|---|---|---|---|---|---|---|
| *Validation Set Baseline* | *0.0150* | *0.0043* | *0.0102* | *0.8301* | *0.0441* | *0.0176* | *0.4889* | *1.0814* | — |
| Transformer | 0.5308 | 0.4410 | 1.2284 | 25.8265 | 0.2324 | 0.2278 | 2.7925 | — | — |
| Transformer (AHo) | 0.4456 | 0.3474 | 0.5351 | 6.4490 | 0.1641 | 0.0593 | 2.3472 | — | — |
| IgLM* [80] | **0.1103** | **0.0484** | **0.0577** | 11.0675 | **0.0413** | 0.0671 | 1.9274 | — | — |
| dyMEAN [44] | **0.1319** | 0.1600 | **0.0423** | 3.9145 | 0.1566 | 0.2929 | 2.3711 | 601.1153 | 3.8157 |
| EGNN | 0.3988 | 0.2655 | 0.3547 | **2.1115** | 0.1486 | 0.1085 | 1.9881 | 1586.0160 | 9.8231 |
| EGNN (AHo) | 0.3329 | 0.2229 | 0.2904 | 8.1620 | 0.1263 | 0.1075 | 0.7978 | 1714.2734 | 10.0628 |
| EGNN (AHo & Cov.) | 0.3482 | 0.2374 | 0.2443 | 2.5632 | 0.1190 | **0.0462** | 1.2184 | 1015.8926 | 9.4814 |
| FA-GNN | 0.4141 | 0.2822 | 0.4302 | 2.5330 | 0.1696 | 0.1164 | 1.7886 | 22.7988 | 0.8617 |
| FA-GNN (AHo) | 0.3407 | 0.2263 | 0.2344 | 2.3272 | 0.1411 | 0.1306 | 1.6046 | **8.7506** | 0.8321 |
| FA-GNN (AHo & Cov.) | 0.2785 | 0.1669 | 0.0815 | 5.4440 | 0.0493 | **0.0212** | 0.7768 | **15.3670** | 0.8814 |
| AbDiffuser (uniform prior) | 0.2837 | 0.1419 | 0.2188 | 3.1364 | 0.0727 | 0.1691 | 1.3874 | 38.8417 | 0.8398 |
| AbDiffuser (no projection) | 0.2378 | 0.1529 | 0.0694 | 2.3530 | 0.0637 | 0.0793 | **0.7376** | 6313.2495 | 11.1431 |
| AbDiffuser (no Cov.) | 0.2309 | 0.1107 | 0.1235 | **1.2392** | 0.0664 | 0.0511 | **0.6453** | 17.7322 | **0.6302** |
| AbDiffuser | 0.1979 | **0.0921** | 0.0662 | **2.3219** | **0.0314** | **0.0285** | **0.6662** | **13.3051** | **0.5230** |
| AbDiffuser (side chains) | **0.0916** | **0.0520** | **0.0186** | 6.3166 | **0.0209** | 0.0754 | 0.8676 | 16.6117 | **0.4962** |

Table 1: Antibody generation based on Paired OAS [61]. AHo denotes models that use AHo numbering and position-specific residue frequencies. Cov denotes models that use the learned covariance. IgLM is denoted by ∗ since it was trained on significantly more data (including the test set) and was given part of the sequence to bootstrap generation. The top three results in each column are highlighted as **First**, **Second**, **Third**.

| Model | Parameters ↑ | Memory (training) ↓ | Memory (generation) ↓ | Generation time ↓ |
|---|---|---|---|---|
| Transformer | 84M | 14GB | 15GB | 3.2 min |
| EGNN | 39.3M | 78GB | 16GB | 22.6 min |
| FA-GNN | 9.4M | 75GB | 38GB | 9.5 min |
| AbDiffuser | 169M | 12GB | 3GB | 2.3 min |

Table 2: Number of parameters, model memory consumption during training with a batch size of 4 and memory consumption with the time taken to generate a batch of 10 examples for paired OAS.

## 5.1 Paired OAS Generation

We focus on matching the distribution of 105k paired sequences from the Observed Antibody Space database [61] folded with IgFold and optimized with Rosetta [2].

Table 1 summarizes the results. Generated samples by AbDiffuser improve upon baselines on nearly all fronts, even compared to the IgLM language model which was trained on magnitudes more data (especially when concerning structure-related metrics). dyMEAN GNN-based model struggles in this distribution modeling task and is the only model tested that does not achieve the perfect uniqueness of the generated samples (58. 2% unique). The experiment also corroborates our analysis (Theorem E.1) on the benefit of informative priors to diffusion: using a position-specific residue type frequency (AHo) and encoding conditional atom dependencies through a learned covariance matrix (Cov.) helps to improve the ability of most models to capture the pOAS distribution. Interestingly, including the learned covariance can sometimes noticeably improve the quality of the generated sequences (FA-GNN), but its strong benefit to structure quality is only felt when the model is powerful enough to model the structure well (APMixer). Inclusion of AHo numbering and position-specific frequencies improves all models. We perform a similar ablation for APMixer by setting the prior distribution to uniform (uniform prior) and observe a similar performance drop.

To interpret the fidelity of the generated structures, we recall that IgFold uses an ensemble of 4 models followed by Rosetta optimization and that, on average, individual IgFold models (before ensembling) achieve an RMSD of $0.4239$ on the test set. Therefore, in this regard, the structures created by AbDiffuser are nearly indistinguishable from the test set structures (RMSD of 0.4962). A more detailed analysis of per-region RMSD can be found in Appendix K. We further observe that when no projection layer is used and instead one uses standard diffusion to predict the noise added to the atom positions [28], the training becomes less stable and the model can fail to learn to generate good structures. Encouragingly, forcing AbDiffuser to model side chain positions alongside the backbone tends to improve the similarity of the generated sequences (Naturalness, Closeness, Stability). This is likely due to the strong coupling between feasible side chain conformations and residue types. The generated side chains get an average Rosetta packing score of 0.624, whereas folded and Rosetta-optimized structures have a mean packing score of 0.666. Recalling that a packing score above 0.6 is widely considered good [77; 66; 3], we deduce that AbDiffuser is able to generate physically plausible side chain conformations. When available, we also use the side chain positions

| Model | $W_1$(Nat.)↓ | $W_1$(Clo.)↓ | $W_1$(Sta.)↓ | $W_1$(PSH)↓ | $W_1$(PPC)↓ | $W_1$(PNC)↓ | $W_1$(CSP)↓ | $W_1(\Delta G)$↓ | RMSD↓ | $p_{\text{bind}}$↑ | Uniq.↑ |
|---|---|---|---|---|---|---|---|---|---|---|---|
| *Validation Set Baseline* | *0.0011* | *0.0003* | *0.0061* | *1.3183* | *0.0196* | *0.0114* | *0.3280* | *2.0350* | *—* | *0.8676* | *100%* |
| MEAN [43] | 0.0072 | **0.0009** | 0.0267 | 8.4184 | 0.0184 | 0.0231 | 0.4108 | 8.5981 | 0.7792 | 0.7767 | 38.9% |
| DiffAb [53] | 0.0074 | 0.0014 | 0.0498 | 0.5481 | 0.0097 | 0.0067 | 3.4647 | **6.7419** | 0.4151 | 0.8876 | 99.7% |
| RefineGNN [34] | 0.0011 | **0.0004** | **0.0026** | 0.5482 | **0.0053** | 0.0046 | **0.1260** | — | — | 0.7132 | 100% |
| Transformer (AHo) | 0.0014 | 0.0031 | 0.0097 | 1.3681 | 0.0171 | 0.0140 | 0.2657 | — | — | 0.3627 | 100% |
| EGNN (AHo & Cov.) | 0.0013 | 0.0030 | 0.0102 | 1.1241 | 0.0123 | 0.0236 | 0.2441 | 1967.5280 | 9.2180 | 0.3626 | 100% |
| FA-GNN (AHo & Cov.) | 0.0018 | 0.0030 | 0.0063 | **0.4158** | **0.0107** | 0.0056 | 0.2644 | 76.7852 | 3.1800 | 0.4576 | 100% |
| AbDiffuser | 0.0013 | 0.0018 | **0.0028** | **0.4968** | 0.0205 | 0.0113 | 0.1588 | **6.4301** | **0.3822** | 0.5761 | 100% |
| AbDiffuser (side chains) | **0.0010** | **0.0005** | 0.0062 | 1.2909 | **0.0115** | **0.0029** | **0.0948** | 32.0464 | 0.4046 | 0.6848 | 100% |
| AbDiffuser ($\tau = 0.75$) | **0.0005** | 0.0011 | 0.0054 | **0.3934** | 0.0148 | 0.0129 | 0.1785 | **6.2468** | **0.3707** | 0.6382 | 100% |
| AbDiffuser (s.c., $\tau = 0.75$) | **0.0005** | **0.0004** | 0.0126 | 1.8510 | 0.0126 | **0.0017** | **0.0917** | 12.8923 | 0.3982 | **0.7796** | 100% |
| AbDiffuser ($\tau = 0.01$) | **0.0008** | 0.0014 | 0.0265 | 2.5944 | 0.0206 | 0.0053 | 0.2378 | 15.2200 | **0.3345** | **0.9115** | 99.7% |
| AbDiffuser (s.c. $\tau = 0.01$) | 0.0015 | 0.0024 | 0.0159 | 1.5043 | 0.0210 | 0.0126 | 0.5173 | 114.4841 | 0.6795 | **0.9436** | 91.4% |

Table 3: Generating Trastuzumab mutants based on the dataset by Mason et al. [55]. The top three results in each column are highlighted as **First**, **Second**, and **Third**. Multiple approaches can generate sequences similar to the test set, but generating predicted binders (large $p_{\text{bind}}$) is considerably harder.

predicted by the model for the $\Delta G$ energy estimation. Even though this procedure is expected to generate results slightly different when compared to backbone-only models (in the latter case missing side chain atoms are first repacked based on the rotamer libraries before the energy minimization step), we still observe a high overlap between the resultant energies. This further highlights the quality of side-chain prediction.

It should be noted that $\Delta G$ energy computation and some additional structure-based metrics [69] (i.e., *CDR PSH*, *CDR PPC*, *CDR PNC*, *SFV CSP*) are inherently susceptible to even minor changes in the geometry of the modeled structures. Thus, in line with the discussions by Raybould et al. [69], the overal trends of these metrics can be used to assess the generated samples as similar or dissimilar to the reference distributions, but one should not fall into the trap of overly focusing on the specific values attained. From this perspective, most structure-based models do sufficiently well on these metrics, perhaps with the exception of EGNN $\Delta G$.

In Table 2 we show that APMixer is able to use an order of magnitude more parameters with a smaller memory footprint during training and offers more efficient sample generation, compared to the baseline architectures, on Nvidia A100 80GB GPU.

## 5.2 Generating HER2 Binders

Antibody generative models can be used to explore a subset of the general antibody space, such as the binders of the target antigen. The purpose of modeling and expanding a set of binders is twofold: a) it allows us to rigorously validate our generative models in a setup more tractable than *denovo* design; b) from a practical standpoint, it sets the ground for the optimization of properties that render binding antibodies drugs, such as developability, immunogenicity, and expression, allowing efficient exploration of the binder space [21; 63]. Note that sufficiently large libraries consisting of antibodies of variable binding affinity are usually discovered during the very early stages of the drug design process by means of high-throughput experiments [64; 76; 55]. Thus, the data used here can be sufficiently easier to obtain than the crystal structures usually assumed in CDR redesign experiments.

We use the Trastuzumab CDR H3 mutant dataset by Mason et al. [55] which was constructed by mutating 10 positions in the CDR H3 of the cancer drug Trastuzumab. The mutants were then evaluated using a high-throughput noisy assay to determine if they bind the HER2 antigen. After discarding all duplicate sequences and sequences that appear in both the binder set and the non-binder set, we are left with 9k binders and 25k non-binders. The generative models were trained only on the binder set. Separately, we train a classifier based on the APMixer architecture to distinguish binders from non-binders. The classifier achieves $87.8\%$ accuracy on a randomly selected test set of 3000 antibodies, implying that the predicted binding probability is a somewhat informative metric for how tightly the binder distribution is approximated by the models.

The computational results summarized in Table 3 evidence that AbDiffuser closely approximates the binder distribution. Although most models achieve results close to those achieved by the validation set[1], the sequence-based baseline struggles to capture the features that convey binding; evidenced by the low predicted binding probability. GNN-based baselines struggle to predict the correct structure

---

[1]The Wasserstein distance of the validation set is likely overestimated here due to the validation set being smaller than the test set (100 vs 1000 samples).

(RMSD above 3) producing structures with unreasonably high Amber energy. As a result, the binding probability of their designs is lower than that of the more powerful structural models.

Next, we look at the two baselines that only redesign CDR H3 instead of generating the whole antibody. RefineGNN [34] manages to closely capture the sequence and biophysical characteristics and generates sufficiently good binders. DiffAb [53], which is another diffusion-based model, achieves the best binding probability out of the CDR redesign baselines. MEAN [43] generated few unique sequences (38.9% uniqueness) and failed some of the distribution-based metrics (Naturalness, CDR PSH). dyMEAN [44] collapsed to always generating a single sample, thus we did not report its results. The overfitting behavior of MEAN and dyMEAN can be attributed to the reconstruction-based training objective and the fact that they were designed for a slightly different task of CDR redesign with different antigens, instead of the single antigen we have here.

Due to the use of iterative refinement and reconstruction-based training objectives, RefineGNN and MEAN focus on the generation of the most likely examples. Focusing on high-likelihood modes is especially beneficial here, as the experiment that was used to create the dataset is noisy. Diffusion models can also be adjusted to focus on prevalent modes by reducing the temperature $\tau$ of the denoising process. In the Gaussian case, we specified the temperature as the scaling factor of the noise added during the reverse process (Appendix B.2), whereas in the discrete case we specified it as the temperature of the model output softmax function (Appendix B.3). We observe that a slight reduction of the temperature helps to improve the general distribution fit across almost all metrics. Reducing the temperature further boosts the binding probability, but, as expected, can result in a slight loss of diversity. Using a higher temperature slightly increases the Stability Wasserstein distance while improving Stability. The phenomenon occurs because the model is no longer concerned with fitting low-likelihood modes of the real distribution that contain structures of poor stability.

In contrast to MEAN, RefineGNN and DiffAb, which only redesign the CDR H3 of the test-set structures, AbDiffuser generates full antibodies and still achieves better sequence similarity. MEAN [43] also produced structures of noticeably worse RMSD, which can be explained as follows: a) as we see in Appendix K, MEAN does not predict CDR H3 positions as well; b) changing CDR H3 can impact the most-likely conformation of the overall antibody, something that the CDR inpainting models cannot account for. We do not report the RMSD and $\Delta G$ for RefineGNN as it does not place the generated CDR H3 loop in the reference frame of the original antibody.

The AbDiffuser model that also generates side chains generally achieved better sequence similarity (Naturalness, Closeness) and better binding probability than the backbone-only model, but a worse similarity of sequence stability. Furthermore, while the side chain model achieved a worse overall structure quality ($\Delta G$, RMSD), as we see in Appendix K it predicted CDR H3 positions more precisely, which is the main desideratum in this experiment.

## 5.3 In Vitro Validation

We further validate our designs through an *in vitro* experiment. As shown in Figure 2, all submitted designs were expressed and purified successfully (average concentration of 1.25 mg/ml) and an average of 37.5% of the designs were confirmed binders with pKD $\in [8.32, 9.50]$ (higher is better) whose average was 8.70. The binding rate was improved (from 22. 2% to 57. 1%) when considering designs that were additionally filtered so that they were better than the bottom 25th quantile in every metric (naturalness, RMSD, etc.) and a classifier trained to distinguish binders from non-binders predicted that they bound with high confidence. This increase in binding for filtered samples suggests that our selected metrics indeed correlate with desirable *in vitro* characteristics. Our best binder belonged to the latter set and its pKD of 9.50 was slightly above Trastuzumab (not in the training set) while differing in 4 positions of the CDR H3 loop. Further examination of the top binders is performed in Figure 3 and Appendix L.

In contrast to the only other comparable study for ML-based Trastuzumab CDR H3 mutant design by Shanehsazzadeh et al. [76], our best binder had pKD $= -\log_{10}(3.17^{-10}) = 9.50$, while the best binder found by Shanehsazzadeh et al. had pKD $= 9.03$. Two important differences between the two studies are that: we trained on binders and tested 16 samples in the wet lab, while Shanehsazzadeh et al. used a simpler sequence infiling model trained on a large set of generic antibodies to generate 440k candidate sequences that were filtered in the wet-lab using high-throughput screening to identify 4k binders, of which 421 were selected to be tested using precise SPR measurements. The fact that our

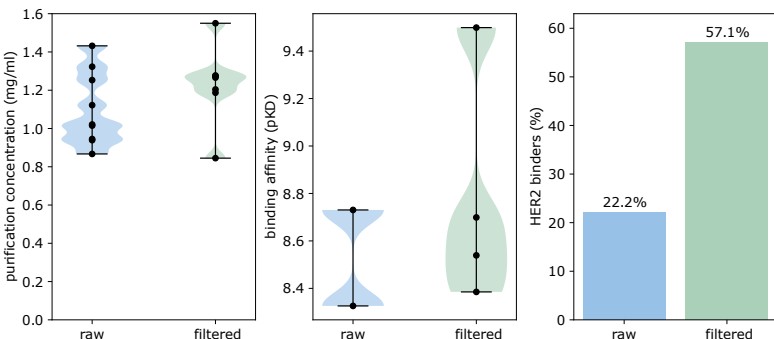

Figure 2: *In vitro* validation of AbDiffuser designs in terms of their ability to express (left), binding affinity (center), and binding rate (right). The 'raw' column corresponds to randomly selected generated antibodies, whereas 'filtered' designs were additionally filtered by *in silico* screening.

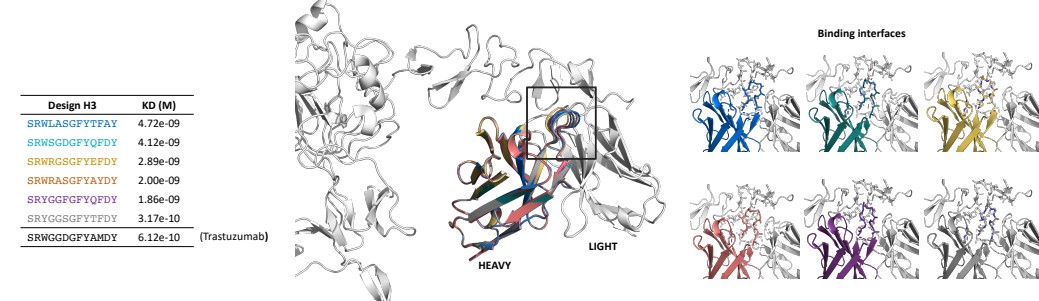

| Design H3 | KD (M) | |
|---|---|---|
| SRWLASGFYTFAY | 4.72e-09 | |
| SRWSGDGFYQFDY | 4.12e-09 | |
| SRWRGSGFYEFDY | 2.89e-09 | |
| SRWRASGFYAYDY | 2.00e-09 | |
| SRYGGFGFYQFDY | 1.86e-09 | |
| SRYGGSGFYTFDY | 3.17e-10 | |
| SRWGGDGFYAMDY | 6.12e-10 | (Trastuzumab) |

Figure 3: HER2 structure and the 3D structures of generated binders highlighted in different colors. AbDiffuser has learned to redesign part of the binding interface while maintaining affinity. Our tightest binder achieved a pKD of 9.5 which, accounting for experimental variability, is akin to Trastuzumab whose average measured pKD was 9.21.

approach required 26x fewer precise SPR wet-lab experiments to find a better binder hints toward a major improvement in efficiency. This highlights the importance of new and more efficient generative models for antibodies and more efficient, and thus powerful architectures such as APMixer.

The work of Shanehsazzadeh et al. [76] also shows that high-throughput wet-lab experiments can be used to build sufficiently large datasets of target-specific binders without starting from a known binder. This provides a straightforward way to train AbDiffuser on new targets.

After good binders are determined through high-throughput and SPR experiments, at the end of the drug discovery process, an experimental 3D structure of the bound complex is often produced. In recent works, redesigning CDRs in such co-crystal structures has been a popular task [53; 43; 34]. To that end, Appendix N investigates how AbDiffuser can be adopted for it and shows that it offers much better sequence recovery rates than the current state-of-the-art diffusion model for the CDR redesign [53].

## 6 Conclusions

We propose a denoising model that deviates from conventional practice in deep learning for protein design. APMixer enjoys the benefits of the best protein models (SE(3) equivariance, universality), while being significantly more memory efficient. We also show how to tailor the diffusion process to antibody generation by incorporating priors that enable the handling of variable-length sequences and allow the model to navigate strong bond constraints while operating in an extrinsic coordinate space that supports Gaussian noise. In our future work, we want to apply our method to protein families beyond antibodies, as organized in CATH [81] and Pfam [57]. Promising examples include the TIM-barrel fold and enzyme families, such as SAM-dependent methyltransferases, long-chain alcohol oxidases, and amine dehydrogenases [83].

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

# A   Related Work

The literature on computational methods for protein design is vast. We focus here on machine learning methods for protein sequence and structure generation and refer the interested reader to the relevant reviews by Woolfson [91]; Ferruz et al. [18] for broader context.

## A.1   Protein Language Models

The first works on protein generation focused on language modeling [16; 21; 71; 85]. Motivated by the observation that the natural forces driving antibody sequence formation, namely V(D)J recombination and the selective pressure in the presence of an antigen referred to as somatic hypermutation, are distinct from other proteins, recent works have also built antibody-specific language models. Therein, large-scale models trained on millions of sequences, such as AntiBERTy [74] and IgLM [80], have been used to generate full-length antibody sequences across different species or restore missing regions in the antibody sequence [62; 56].

## A.2   Protein Structure Generation

These approaches use equivariant architectures to generate proteins and antibodies in 3D. We broadly distinguish two strategies: *backbone-first* and sequence-structure *co-design*.

The *backbone-first* strategy simplifies the generation problem by splitting it into two steps: the generation of the protein backbone and of the protein sequence given the backbone structure. A backbone can be designed by refining a rationally constructed template [24] or by a structure-based generative model [90; 32; 15; 87; 50; 92; 47]. On the other hand, the determination of the residues that fit the backbone can be cast within the self-supervised learning paradigm [3; 11] or by minimizing some score that is a proxy for energy [60]. Despite the promise of backbone-first strategies, in the context of deep learning, it is generally preferable to solve problems in an end-to-end fashion. Decoupling the problem into separate steps can also prohibit the modeling of complex dependencies between backbone and sequence, as early decisions are not updated in light of new evidence. As discussed by Harteveld et al. [24], one failure mode of the backbone-first strategy is obtaining a secondary structure that is not physically realizable with natural amino acids. It is also worth mentioning that structure diffusion generative models can also be used for protein-protein docking, as shown by Ketata et al. [40].

Sequence and structure *co-design* entails learning to generate a full protein in an end-to-end fashion [51; 89]. In this paper we also adopt this strategy. As highlighted in the introduction, AbDiffuser differs from these previous works in the neural architecture and protein representation (APMixer), as well as by shaping the diffusion process to better suit the problem domain.

A relevant line of works has also focused on the re-generation of the complementarity-determining regions (CDRs) of antibodies, a subset of the heavy and light chains containing typically between 48 and 82 residues [10] of particular relevance to binding [34; 20; 43; 33; 53; 78; 44]. These constitute meaningful first steps towards the goal of full-atom antibody design that we consider in this work. Having the ability to regenerate the full antibody sequence and structure significantly enlarges the design space while also enabling the probing of properties beyond affinity, such as stability and immunogenicity. Furthermore, even in the context of affinity optimization, it can be beneficial to also redesign the framework and vernier regions, as the latter sometimes interact with the antigen and affect the CDR conformation [1; 70; 79].

# B   Antibody Diffusion

This section complements the discussion in Section 2 by describing in greater detail how we frame denoising diffusion.

## B.1 Additional Background on Denoising Diffusion

Recall that the forward process $q(X_1, ..., X_T | X_0) = q(X_1|X_0) \Pi_{t=2}^T q(X_t|X_{t-1})$, is Markovian. This Markovianity allows for the recovery of the reverse process

$$q(X_{t-1}|X_t, X_0) = \frac{q(X_t|X_{t-1}) \, q(X_{t-1}|X_0)}{q(X_t|X_0)}.$$

Thus, to generate new data, a neural network learns to approximate the true denoising process $\hat{X}_0 = \phi(X_t, t)$, which can be accomplished by minimizing the variational upper bound on the negative loglikelihood [84; 26; 5]:

$$
\begin{aligned}
L_{\text{elbo}}(X_0) := \mathbb{E}_{q(X_0)} \Bigg[ & \underbrace{D_{KL}(q(X_T|X_0)\|p(X_T))}_{L_T(X_0)} \\
& + \sum_{t=2}^T \mathbb{E}_{q(X_t|X_0)} \underbrace{D_{KL}(q(X_{t-1}|X_t, X_0)\|p_\theta(X_{t-1}|X_t))}_{L_t(X_0)} \\
& - \mathbb{E}_{q(X_1|X_0)} \underbrace{\log(p_\theta(X_0|X_1))}_{L_1(X_0)} \Bigg]
\end{aligned}
\tag{3}
$$

In practice, the model is trained to optimize each of the $L_t(X_0)$ loss terms and the $L_0(X_0)$ loss term individually, by sampling an arbitrary time step $t \in \{1, T\}$. Note that $L_T(X_0)$ approaches 0 for any data distribution if sufficiently many time steps are used. The latent variable distribution $q$ is usually defined so that one can directly sample any step in the trajectory efficiently.

As discussed in Section 2, in our diffusion process, we factorize the posterior probability distribution over the atom positions and residue types:

$$q(X_t|X_{t-1}) = q(X_t^{\text{pos}}|X_{t-1}^{\text{pos}}) \, q(X_t^{\text{res}}|X_{t-1}^{\text{res}}).$$

We discuss the constituent processes in the following two sections.

## B.2 Equivariant Diffusion of Atom Positions

One of the most important features in the training of a diffusion model is the ability to efficiently sample any step in the forward process $q(X_t^{\text{pos}}|X_0^{\text{pos}})$. This is easily achieved if we use a Gaussian distribution $q(X_t^{\text{pos}}|X_0^{\text{pos}}) = \mathcal{N}(\alpha_t X_0^{\text{pos}}, \sigma_t^2 \Sigma)$, where $\alpha_t$ and $\sigma_t^2$ are positive, scalar-valued functions of $t$, with $\alpha_1 \approx 1$ and $\alpha_T \approx 0$. A common way to define the noising process is the variance-preserving parameterization [84; 26] with $\alpha_t = \sqrt{1 - \sigma_t^2}$. A forward process step in this general diffusion formulation can be expressed as $q(X_t^{\text{pos}}|X_{t-1}^{\text{pos}}) = \mathcal{N}(\alpha_{t|t-1} X_0^{\text{pos}}, \sigma_{t|t-1}^2 \Sigma)$, where $\alpha_{t|t-1} = \alpha_t/\alpha_{t-1}$ and $\sigma_{t|t-1}^2 = \sigma_t^2 - \alpha_{t|t-1}^2 \sigma_{t-1}^2$. Similarly, a reverse process step $q(X_{t-1}^{\text{pos}}|X_t^{\text{pos}}, X_0^{\text{pos}})$ can then be expressed as the following Gaussian:

$$\mathcal{N}\left( \frac{\alpha_{t|t-1}\sigma_{t-1}^2 X_t^{\text{pos}} + \alpha_{t-1}\sigma_{t|t-1}^2 X_0^{\text{pos}}}{\sigma_t^2}, \frac{\sigma_{t|t-1}^2 \sigma_{t-1}^2}{\sigma_t^2} \Sigma \right).$$

Similarly to Watson et al. [90], we can reduce the temperature $\tau$ of the Brownian motion of the backward process by scaling the variance of this Gaussian distribution by $\tau$.

Kingma et al. [41] showed that the training objective for the Gaussian distribution can be simplified to

$$L_t(X_0^{\text{pos}}) = \frac{1}{2} \left( \text{SNR}(t-1) - \text{SNR}(t) \right) \|\hat{X}_0^{\text{pos}} - X_0^{\text{pos}}\|_{\Sigma^{-1}}^2,$$

where the signal-to-noise ratio is $\text{SNR}(t) = (\alpha_t/\sigma_t)^2$ and $\|\cdot\|_{\Sigma^{-1}}^2 = \text{tr}\left( (\cdot)^T \Sigma^{-1}(\cdot) \right)$.

It is quite common in the literature to choose a slightly different optimization objective. In particular, it has been found that predicting the noise $\hat{\epsilon} = \phi(X_t^{\text{pos}}, t)$ that has been added $X_t^{\text{pos}} = \alpha_t X_0^{\text{pos}} + \sigma_t \epsilon$,

$\epsilon \sim \mathcal{N}(0, \Sigma)$ and using an unweighted loss $L_t(X_0^{\text{pos}}) = \frac{1}{2}\|\hat{\epsilon} - \epsilon\|_{\Sigma^{-1}}^2$ improves model training [26; 41; 28]. A possible explanation is that this parameterization makes it easier for the model to minimize the loss at time steps close to $T$. We found that a similar effect can be obtained by re-weighing the mean prediction loss (Appendix C):

$$L_t(X_0^{\text{pos}}) = \frac{1}{2}\, \text{SNR}(t)\, \|\hat{X}_0^{\text{pos}} - X_0^{\text{pos}}\|_{\Sigma^{-1}}^2, \tag{4}$$

which empirically renders the mean $\hat{X}_0^{\text{pos}} = \phi(X_t^{\text{pos}}, t)$ and error $\hat{\epsilon} = \phi(X_t^{\text{pos}}, t)$ prediction alternatives perform comparably. We rely on equation 4 throughout our experiments, since predicting $\hat{X}_0^{\text{pos}} = \phi(X_t^{\text{pos}}, t)$ allows for easier incorporation of known constraints, such as atom bond lengths (see Section 3.4). As shown in Appendix D, $L_1(X_0^{\text{pos}})$ can also be optimized using the same objective.

### B.3 Discrete Residue-type Diffusion

To define the discrete diffusion process for the residue types we adopt the formulation introduced by Austin et al. [5]. If we have a discrete random variable with $k$ categories, represented as a one-hot vector $X^{\text{res}}$, we define the forward process $q(X_t^{\text{res}}|X_0^{\text{res}}) = \text{Cat}(X_0^{\text{res}}Q_t)$, where the transition matrix $Q_t = \beta_t I + (1 - \beta_t)Q$ is a linear interpolation between identity and target distribution $Q$ (e.g., uniform distribution). Similarly to atom diffusion, one forward process step is $q(X_t^{\text{res}}|X_{t-1}^{\text{res}}) = \text{Cat}(X_t^{\text{res}}Q_{t|t-1})$, where $Q_{t|t-1} = Q_{t-1}^{-1}Q_t$. From this, a single reverse process step can be expressed as:

$$q(X_{t-1}^{\text{res}}|X_t^{\text{res}}, X_0^{\text{res}}) = \text{Cat}\left(\frac{X_t^{\text{res}}Q_{t|t-1}^T \odot X_0^{\text{res}}Q_{t-1}}{\mathcal{Z}}\right),$$

for an appropriate normalizing constant $\mathcal{Z}$. In the discrete case, we implement temperature $\tau$ scaling as the temperature of the softmax applied to the model output. This serves to sharpen the target categorical distribution $\sim p^{1/\tau}$.

Similarly to the atom diffusion case, we use a simplified objective for the discrete diffusion [22; 7] which has been found to lead to better results:

$$L_t(X_0^{\text{res}}) = -\beta_t \log p_\theta(X_0^{\text{res}}|X_t^{\text{res}})$$

Hyperparameter $\beta_t$ ensures that the loss weight is proportional to the fraction of unchanged classes and we set $\beta_t = \alpha_t^2$ to keep the noise schedules for the residue-type diffusion and atom-position diffusion the same. We also note that now the $L_1(X_0)$ loss term is simply the cross-entropy of the model prediction $L_1(X_0) = \log p_\theta(X_0|X_1)$. The derivation of our loss can be found in Appendix E.

## C  Re-weighted Optimization Objective for Atom Positions

As shown by Kingma et al. [41], the $L_t$ term in the variational upper bound corresponds to

$$L_t(X_0^{\text{pos}}) = \frac{1}{2}\left(\text{SNR}(t-1) - \text{SNR}(t)\right)\|\hat{X}_0^{\text{pos}} - X_0^{\text{pos}}\|_{\Sigma^{-1}}^2.$$

If we instead parameterize $X_t^{\text{pos}} = \alpha_t X_0^{\text{pos}} + \sigma_t \epsilon$ we can directly re-write this as

$$L_t(\epsilon) = \frac{1}{2}\left(\frac{\text{SNR}(t-1)}{\text{SNR}(t)} - 1\right)\|\hat{\epsilon} - \epsilon\|_{\Sigma^{-1}}^2.$$

Meanwhile, the simplified objective of Ho et al. [26] is instead defined as

$$L_t(\epsilon) = \frac{1}{2}\|\hat{\epsilon} - \epsilon\|_{\Sigma^{-1}}^2.$$

If we reverse the parameterization as $\epsilon = \frac{X_t^{\text{pos}} - \alpha_t X_0^{\text{pos}}}{\sigma_t}$ and using the fact that $\text{SNR}(t) = \alpha_t^2/\sigma_t^2$ this becomes

$$L_t(X_0^{\text{pos}}) = \frac{1}{2}\left\|\frac{X_t^{\text{pos}} - \alpha_t \hat{X}_0^{\text{pos}}}{\sigma_t} - \frac{X_t^{\text{pos}} - \alpha_t X_0^{\text{pos}}}{\sigma_t}\right\|_{\Sigma^{-1}}^2 = \frac{1}{2}\,\text{SNR}(t)\,\|\hat{X}_0^{\text{pos}} - X_0^{\text{pos}}\|_{\Sigma^{-1}}^2.$$

## D Reconstruction Loss for Atom Positions

For the Gaussian atom position diffusion (Section B.2 we need to account for a few considerations in the reconstruction term $L_1(X_0)$ of the variational upper bound. Assuming that the data probability is constant, $q(X_0^{\text{pos}}|X_1^{\text{pos}}) \approx \mathcal{N}(\frac{X_1^{\text{pos}}}{\alpha_1}, \frac{\sigma_1^2}{\alpha_1^2}\Sigma)$. Similarly to Hoogeboom et al. [28], we parameterize the mean using the model prediction instead:

$$L_1(X_0^{\text{pos}}) = \log \mathcal{Z}^{-1} - \frac{1}{2}\text{SNR}(1)\|X_0^{\text{pos}} - \hat{X}_0^{\text{pos}}\|_{\Sigma^{-1}}^2.$$

As highlighted by Hoogeboom et al. [28], this increases the quality of the results.

Isotropic distributions, such as Gaussians with appropriate covariance matrices, assign the same probability to every rotation of a vector in 3D, implying that the resulting atom diffusion process is equivariant w.r.t. orthogonal group O3. To additionally impose that the diffusion is translation-equivariant, we follow Hoogeboom et al. [28] and use a re-normalized multivariate Gaussian distribution, where the sample center of mass is fixed to zero: $\sum_{i=1}^{n} X_i^{\text{pos}} = 0$. The latter has the same law as an ordinary Gaussian with the main difference that the normalizing constant is now $\mathcal{Z} = \left(\sqrt{2\pi}\sigma\right)^{3(n-1)}$ for $n$ atoms in 3D space.

It is clear to see that the O3 property is satisfied if we have an identity covariance. However, it also holds for non-isotropic covariance matrices. Let us define $Z \sim \mathcal{N}(0, I)$ of dimension $n \times 3$ and $X^{\text{pos}} = KZ$, where $\Sigma^{-1} = KK^T$. The latter can be re-written as $x_0 = \text{vec}(X_0) = (I \otimes K)\text{vec}(Z) = \Sigma^{1/2}z$. We then note that for any 3x3 unitary matrix $U$, we have $x_0^\top (U^\top \otimes I)\Sigma^{-1}(U \otimes I)x_0 = x_0^\top (U^\top \otimes I)(I \otimes L)(U \otimes I)x_0 = x_0^\top (U^\top U \otimes L)x_0 = x_0^\top (I \otimes L)x_0$. In other words, since the covariance acts on the node dimension and the unitary matrix acts on the spatial dimension, they are orthogonal and do not affect each other.

## E Simplified Discrete Diffusion Optimization Objective

We can obtain the simplified training objective by taking a slightly different bound on the negative log-likelihood to the one presented in Equation 3 [22]:

$$-\log(p_\theta(X_0)) \leq \mathbb{E}_{X_{1:T} \sim q(X_{1:T}|X_0)}\left[-\log\left(\frac{p_\theta(X_{0:T})}{q(X_{1:T}|X_0)}\right)\right]$$

$$= \mathbb{E}_q\left[-\log\left(\frac{p_\theta(X_{0:T})}{q(X_{1:T}|X_0)}\right)\right]$$

$$= \mathbb{E}_q\left[-\log(p_\theta(X_T)) - \sum_{t=2}^{T}\log\left(\frac{p_\theta(X_{t-1}|X_t)}{q(X_t|X_{t-1})}\right) - \log\left(\frac{p_\theta(X_0|X_1)}{q(X_1|X_0)}\right)\right]$$

$$= \mathbb{E}_q\left[-\log(p_\theta(X_T)) - \log\left(\frac{p_\theta(X_0|X_1)}{q(X_1|X_0)}\right)\right.$$

$$\left. - \sum_{t=2}^{T}\log\left(\frac{q(X_{t-1}|X_t, X_0)\, p_\theta(X_0|X_t)}{q(X_{t-1}|X_t, X_0)} \cdot \frac{q(X_{t-1}|X_0)}{q(X_t|X_0)}\right)\right]$$

$$= \mathbb{E}_q\left[\underbrace{-\log\left(\frac{p_\theta(X_T)}{q(X_T|X_0)}\right)}_{L_T(X_0)} - \sum_{t=2}^{T}\underbrace{\log(p_\theta(X_0|X_t))}_{L_t(X_0)} - \underbrace{\log(p_\theta(X_0|X_1))}_{L_1(X_0)}\right]$$

In this formulation, compared to equation 3, the term $L_t = D_{\text{KL}}(q(X_{t-1}|X_t, X_0)\,\|p_\theta(X_{t-1}|X_t))$ is replaced by

$$L_t = -\log(p_\theta(X_0|X_t))$$

To form the discrete loss $L_t(X_0^{\text{res}})$, we additionally weight each loss term $L_t$ by $\beta_t$ to match the loss weighting used for the atom positions (Appendix C). Note that when this objective is minimized, the original $D_{\text{KL}}$ term will also be at a minimum.

## E.1 Non-Informative Priors Demand Higher Complexity Denoising Models

Denote by $p$ the data distribution and by $p_\theta$ the learned distribution, the latter of which is obtained by the push-forward $p_\theta(f_\theta(Z)) = q(Z)$ of some prior measure $q$ by function $f_\theta$. In the context of denoising diffusion, $Z = X_T$, while $f_\theta = \phi(X_T, T)$ is the learned reverse process.

We prove the following general result which relates the informativeness of the prior with the quality of the generative model and the complexity of the function computed:

**Theorem E.1.** *For any $f_\theta$ that is an invertible equivariant function w.r.t. a subgroup $G$ of the general-linear group GL(d,$\mathbb{R}$) the following must hold:*

$$c_q(f_\theta) \geq W_t(p, q) - W_t(p, p_\theta),$$

*where $c_q(f) := (\min_{g \in G} \mathbf{E}_{Z \sim q}[\|f(g\,Z) - Z\|_t^t])^{1/t}$ quantifies the expected complexity of the learned model under $q$, $W_t(p, p_\theta)$ is the Wasserstein $t$-distance of our generative model to the data distribution, and $W_t(p, q)$ is the distance between prior and posterior.*

The bound asserts that one cannot learn a simple (in terms of $c_q(f)$) generative model that fits the data well unless the prior has a small Wasserstein distance to the data distribution.

The complexity measure $c_q(f)$ is particularly intuitive in the context of denoising diffusion. For the diffusion of atom positions with a SE(3) equivariant model $\phi$, we have

$$c_q(\phi) = \min_{g \in \text{SE(3)}} \mathbf{E}_{X_T}[\|\phi(g\,X_T, T) - X_T\|_2^2],$$

that corresponds to the expected amount of denoising that the model performs throughout the reverse process, discounting for rotations. By selecting an informative prior, we decrease $W_t(p, q)$ and therefore reduce the amount of denoising our model needs to do.

*Proof.* We recall that the Wasserstein $t$-distance between two probability measures $p, p'$ with finite $t$-moments is

$$W_t(p, p') := \left( \inf_{\gamma \in \Gamma(p, p')} \mathbf{E}_{(X, X') \sim \gamma}[\|X - X'\|_t] \right)^{1/t},$$

where the coupling $\gamma$ is a joint measure on $\mathbb{R}^d \times \mathbb{R}^d$ whose marginals satisfy $\int_{\mathbb{R}^d} \gamma(X, X')dx' = p(X)$ and $\int_{\mathbb{R}^d} \gamma(X, X')dx = p'(X')$, whereas $\Gamma(p, p')$ is the set of all couplings of $p, p'$.

We will first lower bound $W_1(p_\theta, q)$. To achieve this, let $X = f(Z)$ and $X' = f_\theta(Z')$ and select $\gamma^*(X, X') = 0$ when $Z \neq Z'$ and $\gamma^*(X, X') = q(Z)$ otherwise, which is a coupling because it satisfies $\int_{X'} \gamma^*(X, X')dx' = q(Z) = p(X)$ and $\int_X \gamma^*(X, X')dx = q(Z') = p_\theta(X')$. Then, exploiting that the Wasserstein distance is a metric (and thus abides by the triangle inequality), we get

$$
\begin{aligned}
W_t(p_\theta, q) &\geq W_t(p, q) - W_t(p, p_\theta) && \text{(by the triangle inequality)} \\
&= W_t(p, q) - \left( \inf_{\gamma \in \Gamma(p, p_\theta)} \int_{X, X'} \gamma(X, X') \|X - X'\|_t^t \, dx \, dx' \right)^{1/t} \\
&\geq W_t(p, q) - \left( \int_{X, X'} \gamma^*(X, X') \|X - X'\|_t^t \, dx \, dx' \right)^{1/t} \\
&= W_t(p, q) - \left( \int_X \left( \int_{X'} \gamma^*(X, X') \|X - X'\|_t^t \, dx' \right) dx \right)^{1/t} \\
&= W_t(p, q) - \left( \int_X p(X) \|X - f_\theta(f^{-1}(X))\|_t^t dx \right)^{1/t} \\
& && \text{(by the definition of } \gamma^* \text{ and invertibility of } f) \\
&= W_t(p, q) - \left( \mathbf{E}_{X \sim p}[\|X - f_\theta(f^{-1}(X))\|_t^t] \right)^{1/t} \\
&= W_t(p, q) - \left( \mathbf{E}_{Z \sim q}[\|f(Z) - f_\theta(f^{-1}(f(Z)))\|_t^t] \right)^{1/t} \\
&= W_t(p, q) - \left( \mathbf{E}_{Z \sim q}[\|f(Z) - f_\theta(Z)\|_t^t] \right)^{1/t} && (5)
\end{aligned}
$$

Next, we upper bound $W_t(p_\theta, q)$. Let $G$ be any subgroup of the general linear group $GL(d, \mathbb{R})$ for which $f_\theta(Z)$ is equivariant and further denote by $g \in G$ an element of the said group.

We fix $\gamma^\#(X, Z) = q(Z)$ when $X = f_\theta(gZ)$ and $\gamma^\#(X, Z) = 0$, otherwise. To see that the latter is a valid coupling notice that $f_\theta g$ is bijective as the composition of two bijective functions, implying:

$$\int \gamma^\#(X, Z) dz = q(z_x) \quad \text{and} \quad \int \gamma^\#(X, Z) dx = q(Z),$$

with $z_x = (f_\theta \circ g)^{-1}(X)$. Further,

$$
\begin{aligned}
q(z_x) &= p_\theta(f_\theta(g^{-1}(f_\theta^{-1}(X)))) && \text{(since } p_\theta(X) = q(f_\theta(Z))\text{)} \\
&= p_\theta(g^{-1} f_\theta(f_\theta^{-1}(X))) && \text{(by the equivariance of } f_\theta\text{)} \\
&= p_\theta(g^{-1} X) \\
&= p_\theta(X), && \text{(by the exchangeability of } p_\theta\text{)}
\end{aligned}
$$

as required for $\gamma^\#$ to be a coupling. We continue by bounding the Wasserstein metric in light of $\gamma^\#$:

$$
\begin{aligned}
W_t(p_\theta, q)^t &= \inf_{\gamma \in \Gamma(p_\theta, q)} \int \gamma(X, Z) \|X - Z\|_t^t \, dx \, dz && \text{(by definition)} \\
&\leq \int \gamma^\#(X, Z) \|X - Z\|_t^t \, dx \, dz && \text{(due to the inifimum)} \\
&= \int \left( \int \gamma^\#(X, Z) \|X - Z\|_t^t \, dx \right) dz \\
&= \int q(Z) \|f_\theta(gZ) - Z\|_t^t \, dz && \text{(by definition of } \gamma^\#\text{)} \\
&= \mathbf{E}_{Z \sim q}[\|f_\theta(gZ) - Z\|_t^t] && (6)
\end{aligned}
$$

Combining equation 5 and equation 6, we get

$$\min_{g \in G} \mathbf{E}_{Z \sim q}[\|f_\theta(gZ) - Z\|_t^t] \geq W_t(p, q) - \left( \mathbf{E}_{Z \sim q}[\|f(Z) - f_\theta(Z)\|_t^t] \right)^{1/t}, \quad (7)$$

as claimed.

In the context of our paper, we have $Z = X_T \sim \mathcal{N}(0, \Sigma)$, $f_\theta(Z) = \hat{X}_0 = \phi(X_T, T)$, $f(Z) = X_0$, $G$ is the standard Euclidean group SE(3), and we select $t = 2$. Then, the aforementioned inequality becomes

$$\min_{g \in \text{SE}(3)} \mathbf{E}_{X_T}[\|\phi(X_T, T) - g X_T\|_2^2] \geq W_2(p, q_T) - \sqrt{\mathbf{E}_{X_T}[\|\phi(X_T, T) - \phi^*(X_T, T)\|_2^2]}, \quad (8)$$

where we have changed the position of $g$ by exploiting that the $\ell_2$ norm is unitary invariant and that $\phi$ is SE(3) equivariant. Alternatively, we can obtain the following tighter bound by stopping earlier at the derivation of equation 5:

$$\mathbf{E}_{X_T}[\|\phi(X_T, T) - X_T\|_2^2] \geq W_2(p, q_T) - W_2(p_\theta, p), \quad (9)$$

where $q_T$ is the law of $X_T$.

$\square$

## F  Learning the Conditional Dependence Relations Between Atoms

We consider the zero-mean multivariate Gaussian whose inverse covariance matrix $\Sigma^{-1} = L + aI$ corresponds to a sparse weighted Laplacian matrix $L$. Intuitively, $L_{ij}$ should be zero if the position of the point $i$ is independent of that of $j$ given the positions of the neighbors of $i$ being fixed. The term $a$ is a small constant whose role is to shift the spectrum of $L$ and thus make the Laplacian invertible.

We follow the framework of [39] and optimize $L$ by minimizing the following objective:

$$\min_{L \in \mathcal{L}} \text{tr}((X^{\text{pos}})^T L X^{\text{pos}}) + f(L),$$

where the $m \times 3$ matrix $X$ contains the 3D positions of the $m$ points modeled and $f(L)$ are some optional constraints we can enforce on the graph Laplacian (and thus also to the adjacency matrix $A$).

The log-likelihood of vectors $X$ on the graph defined by $L$ can be directly expressed as a function of the graph adjacency matrix and the pairwise distance matrix $Z$:

$$\mathrm{tr}((X^{\mathrm{pos}})^T L X^{\mathrm{pos}}) = A \odot Z = \frac{1}{2} \sum_{i,j} A_{ij} \|X_i^{\mathrm{pos}} - X_j^{\mathrm{pos}}\|_2^2.$$

Kalofolias [39] propose the following optimization objective, to recover the optimal adjacency, which with a logarithmic term also ensures that every node has at least one neighbor, which is something we also desire:

$$\min_{A \in \mathcal{A}} \|A \odot Z\|_1 - \mathbf{1}^T \log(A\mathbf{1}) + \frac{1}{2}\|A\|_2^2.$$

An efficient primal-dual algorithm exists for solving this optimization problem [39]. The adjacency matrix that we recover using the mean distance matrix over all of the proteins in paired OAS can be seen in Figure 4. Our use of AHo numbering allows us to directly estimate such mean distances, as with our representation every antibody effectively possesses the same number of residues.

To increase the stability of the Cholesky decomposition $L = KK^T$, which is needed for sampling co-variate noise $X_t^{\mathrm{pos}} = \alpha_t X_0^{\mathrm{pos}} + \sigma_t K\epsilon$, $\epsilon \sim \mathcal{N}(0,1)$, we instead use the generalized Laplacian $L = \mathrm{diag}(A\mathbf{1}) - A + I$ for the diffusion process.

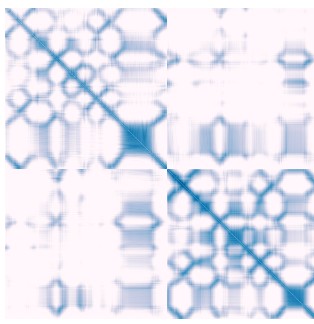

Figure 4: We define a Gaussian prior on the atom positions by learning an adjacency (conditional dependence) matrix for antibody backbone atom positions from all of the folded paired heavy and light chains in the Observed Antibody Space (OAS) [61]. Dependencies between framework residue atom positions and even correlations between heavy (top-left) and light (bottom-right) chain atom positions are distinctly captured.

## G    Proof of SE(3) universality

In the following, we prove the SE(3) universality of an MLP-mixer backbone model combined with frame-averaging.

Concretely, let $X$ be an $n \times m$ matrix with bounded entries and consider the backbone model: $\phi(X) = c_L \circ r_{L-1} \circ \cdots \circ c_3 \circ r_2 \circ c_1(X)$, where the subscript denotes layer, $c_l$ is an MLP operating independently on each column of its input matrix, and similarly, $r_l$ is an MLP operating independently on each row of its input.

It is a consequence of Theorem 4 in [68] that if $\mathcal{F}$ is a bounded $G$-equivariant frame and if $\phi$ is a backbone model that is universal over any compact set, then combining $\phi$ with frame-averaging over a frame-finite domain (i.e., a domain within which the frame cardinality is finite) yields a $G$-equivariant universal model $\langle \phi \rangle_\mathcal{F}$. Since the SE(3) frame is bounded and thus the domain is frame-finite (see also Proposition 1 [68]), to show that $\langle \phi \rangle_\mathcal{F}$ is a universal approximator of any continuous SE(3) function, it suffices to show that the MLP-mixer is a universal approximator over continuous functions over any compact set (or simply universal).

To proceed, we will show that there exists a parameterization so that $\phi(X) = c_L([z; \mathrm{vec}(X)])$ for some vector $z$. We can then rely on the universal approximation property [23] of MLPs to conclude that $\phi$ is also universal.

We first provide an exact constructive argument on how this can be achieved:

1. Let $V$ be a $(n+1) \times (n+1)$ unitary matrix whose first column is the constant vector $\mathbf{1}$ and set $U = V_{:,1:}$ to be the $n+1 \times n$ submatrix. The first MLP embeds each column vector $x$ isometrically into $n+1$ as follows: $c_1(x) = Ux$. By construction, every column of the output $X'$ is now orthogonal to $\mathbf{1}$.

2. The network appends $U$ to the left of $X'$ and $(\overbrace{UX; \cdots ; UX}^{\times n-1})$ to the right. Since appending to the right involves replicating each row multiple times, it can be done by a linear row operation. Appending to the left is slightly more involved and can be achieved by iteratively building $U$ column by column, with each column added by using a row MLP to add one dimension with a constant value and a column MLP to project the new constant vector introduced to the appropriate column of $U$. The key here is that, since we have guaranteed that the columns are always orthogonal to the constant vector, the column MLP can always distinguish the newly introduced column from the ones already at play. Let $X''$ be the result of this process.

3. A column MLP projects each column of the $(n+1) \times (n+nm)$ matrix to $n$ dimensions by multiplying by $U^\top$. The output of this layer is $X''' = [I; X; \cdots ; X]$.

4. A row MLP zeros out all elements except those indexed by the one-hot encoding in the first $n$ entries. The latter can be done by a ReLU MLP with one hidden layer: if all entries of our input are smaller than $b$ and the input is $(\text{onehot}(i), x)$, the first linear layer adds $b$ to only the entries $[n+in : n+in+n)$ and shifts every element by $-b$ using its bias. The latter lets the correct elements pass the ReLU with an unchanged value while setting all other elements to zero, as required. Let the output of this layer be $X''''$ with
$$X''''_{i,:} = (I_{i,:}; \overbrace{0; \cdots 0}^{\times i-1}; X_{i,:}; \overbrace{0; \cdots 0}^{\times n-i}).$$

5. A column MLP sums up all entries. The output is now a row vector $(\mathbf{1}; X_{1,:}^\top; \cdots ; X_{n,:}^\top)^\top$ of length $n + nm$, as required.

Steps 1 and 2 of the above *constructive* argument are inefficient because they rely on very simple layers. This can be avoided if we utilize more powerful MLPs (i.e., functions $c_l$ and $r_l$) and an *existence* argument. Specifically, as there exists a continuous function that achieves step 2, by the universality of MLPs there also exists an MLP that approximates the step arbitrarily well. Similarly, rather than increasing the dimension to $n+1$ at step 1, one may argue that the MLP can assign some vector outside of the bounded domain where the input data resides, and thus the dimension can remain $n$. Therefore, in this setting, only a constant number of MLPMixer blocks are needed to obtain the desired output.

This concludes the proof.

## H   Noise Schedule

In our denoising model, we use the cosine-like schedule proposed by Hoogeboom et al. [28], with a 1000 denoising steps:

$$\alpha_t = (1 - 2s) \cdot \left(1 - \left(\frac{t}{T}\right)^2\right) + s,$$

where a precision $s = 1e^{-4}$ is used to avoid numerical instability. As $\sigma_t = \sqrt{1 - \alpha_t^2}$ and $\beta_t = \alpha_t^2$, it is sufficient to define just $\alpha_t$. Following Nichol and Dhariwal [59] we clip $\alpha_{t|t-1}$ to be at least 0.001, to avoid instability during sampling and then recompute $\alpha_t$ as a cumulative product. To further ensure numerical stability, as recommended by Kingma et al. [41] we compute the negative log SNR curve $\gamma(t) = -\log \alpha_t^2 + \log \sigma_t^2$. This parameterization allows for numerically stable computation of all important values [41], for example, $\alpha_t^2 = \text{sigmoid}(-\gamma(t))$, $\sigma_t^2 = \text{sigmoid}(\gamma(t))$, $\sigma_{t|t-1}^2 = -\text{expm1}(\text{softplus}(\gamma(t-1)) - \text{softplus}(\gamma(t))) = -\text{expm1}(\gamma(t-1) - \gamma(t))\sigma_t^2$, and $\text{SNR} = \exp(-\gamma(t))$. Where $\text{expm1}(x) = \exp(x) - 1$ and $\text{softplus}(x) = \log(1 + \exp(x))$.

# I Full Training and Sampling Algorithms

For convenience, in Algorithms 1 and 2 we describe the full training and sampling procedures.

---

**Algorithm 1** Training

---

1: **repeat**
2: *Get a data point*:
3: $X_0^{\text{pos}}, X_0^{\text{seq}} \sim q(X_0^{\text{pos}}, X_0^{\text{seq}})$
4: *Sample some time step*:
5: $t \sim \text{Uniform}(1, ..., T)$
6: $X_t^{\text{pos}} \sim \mathcal{N}(\alpha_t X_0^{\text{pos}}, \sigma_t^2 \Sigma)$
7: $X_t^{\text{seq}} \sim \text{Categorical}(X_0^{\text{res}} Q_t)$, *here* $Q_t = \beta_t I + (1 - \beta_t) Q$
8: *Predict*:
9: $\hat{X}_0^{\text{pos}}, \hat{X}_0^{\text{seq}} = f_\theta \left( \text{Project}\left(X_t^{\text{pos}}\right), X_t^{\text{seq}}, t \right)$
10: $\hat{X}_0^{\text{pos}} = \text{Project}\left( \hat{X}_0^{\text{pos}} \right)$
11: *Compute losses*:
12: $L_t(\hat{X}_0^{\text{pos}}) = \frac{1}{2} \text{SNR}(t) \|\hat{X}_0^{\text{pos}} - X_0^{\text{pos}}\|_{\Sigma^{-1}}^2$
13: $L_t(\hat{X}_0^{\text{res}}) = \beta_t \text{CrossEntropy}\left( \hat{X}_0^{\text{res}}, X_0^{\text{res}} \right)$
14: *Take a gradient step on*: $\nabla_\theta \left( L_t(\hat{X}_0^{\text{pos}}) + L_t(\hat{X}_0^{\text{seq}}) \right)$
15: **until** converged

---

**Algorithm 2** Sampling

---

1: *Initialize*:
2: $X_T^{\text{pos}} \sim \mathcal{N}(0, \Sigma)$
3: $X_T^{\text{seq}} \sim \text{Categorical}(Q)$
4: *Loop over the time steps*:
5: **for** $t = T, ..., 1$ **do**
6: *Predict*:
7: $\quad \hat{X}_0^{\text{pos}}, \hat{X}_0^{\text{seq}} = f_\theta \left( \text{Project}\left(X_t^{\text{pos}}\right), X_t^{\text{seq}}, t \right)$
8: $\quad \hat{X}_0^{\text{pos}} = \text{Project}\left( \hat{X}_0^{\text{pos}} \right)$
9: *Sample next state*:
10: $\quad$ **if** $t > 1$ **then**
11: $\quad\quad X_{t-1}^{\text{pos}} \sim \mathcal{N}\left( \frac{\alpha_{t|t-1} \sigma_{t-1}^2 X_t^{\text{pos}} + \alpha_{t-1} \sigma_{t|t-1}^2 \hat{X}_0^{\text{pos}}}{\sigma_t^2}, \frac{\sigma_{t|t-1} \sigma_{t-1}}{\sigma_t} \Sigma \right)$
12: $\quad\quad X_{t-1}^{\text{seq}} \sim \text{Categorical}\left( \frac{X_t^{\text{res}} Q_{t|t-1}^T \odot \hat{X}_0^{\text{res}} Q_{t-1}}{\mathcal{Z}} \right)$
13: $\quad$ **else**
14: $\quad\quad X_0^{\text{pos}} \sim \mathcal{N}(\hat{X}_0^{\text{pos}}, \frac{\sigma_1^2}{\alpha_1^2} \Sigma)$
15: $\quad\quad X_0^{\text{seq}} \sim \text{Categorical}\left( \hat{X}_0^{\text{seq}} \right)$
16: **return** $X_0^{\text{pos}}, X_0^{\text{seq}}$.

---

# J Metrics Used in the Numerical Experiments

The metrics applied to generated sequences include methods that measure naturalness, similarity to the closest extant antibody, and stability. To examine *Naturalness* we use the inverse perplexity of the AntiBERTy [74] model trained on a large corpus of antibody chain sequences. As shown by Bachas et al. [6] this score correlates with antibody developability and immunogenicity. To explore *Closeness* we use fast sequence alignment [9] to determine the closest sequence in the training set. The mean sequence identity (fractional edit distance) to the closest training sequence is then reported as a score. Here *Stability* is operationalized by estimating the error of the folded structure using IgFold [73]. This error can be used to rank the sequences by how stable we expect their folds to

be. We take the 90th percentile of the residue error as an estimate of the sequence fold stability; the latter typically corresponds to the CDR H3 and L3 loops, which have the most influence over the antibody's functional properties. To further verify that the generated antibody sequences satisfy appropriate biophysical properties, we rely on four additional structure-based metrics Raybould et al. [69]: CDR region hydrophobicity (*CDR PSH*), patches of positive (CDR PPC), and negative charge (*CDR PNC*), and symmetry of electrostatic charges of heavy and light chains (*SFV CSP*). These metrics operate on folded sequences (using IgFold) and take into account distance-aggregated structural properties of CDR regions (and their spatial vicinity), and their significant deviation from reference values is typically associated with bad developability properties of antibodies such as poor expression, aggregation, or non-specific binding [69].

To evaluate the overall similarity of the generated and the test-set distributions, for all of these sequence metrics, we report the Wasserstein distance between the scores of the sequences in the test split and the scores of the generated sequences.

The metrics applied to the generated structures focus primarily on scores known to correlate with free energy and RMSD. a) We use free energy $\Delta G$ estimated using Rosetta [2] to evaluate the stability of the generated structure. Although lower energy is normally associated with higher stability of the protein structure, one needs to be careful not to reward disproportionly small energies achieved when a miss-formed protein collapses into a morph-less aggregate. Thus, we again report the Wasserstein distance between the generated and test energy distributions. b) *RMSD.* We also re-fold the generated sequence with IgFold [73] and report the RMSD for the backbone $N, C_\alpha, C, C_\beta$ and $O$ atom positions. RMSD is reported as a mean over the generated structures as it captures how well each generated structure matches its sequence.

# K  Detailed Generated Structure RMSD Evaluation

| Model | Full ↓ | Fr ↓ | Fr. H ↓ | CDR H1↓ | CDR H2↓ | CDR H3↓ | Fr. L ↓ | CDR L1↓ | CDR L2↓ | CDR L3↓ |
|---|---|---|---|---|---|---|---|---|---|---|
| EGNN | 9.8231 | 9.3710 | 9.2929 | 13.1720 | 13.0032 | 10.3360 | 9.3918 | 14.6768 | 10.1584 | 10.4860 |
| EGNN (AHo) | 10.0628 | 9.4717 | 9.3552 | 13.1730 | 13.4611 | 12.2434 | 9.5314 | 15.3884 | 10.6975 | 11.0732 |
| EGNN (AHo & Cov.) | 9.4814 | 8.7581 | 8.6206 | 12.9454 | 13.2237 | 12.0939 | 8.8174 | 15.2841 | 10.0504 | 11.1167 |
| FA-GNN | 0.8617 | 0.5748 | 0.5093 | 0.6671 | 0.7438 | 2.2530 | 0.6157 | 0.8199 | 0.5946 | 1.1576 |
| FA-GNN (AHo) | 0.8321 | 0.4777 | 0.4618 | 0.6881 | 0.7867 | 2.2884 | 0.4860 | 0.9398 | 0.5053 | 1.1165 |
| FA-GNN (AHo & Cov.) | 0.8814 | 0.5934 | 0.5236 | 0.5968 | 0.6213 | 2.0788 | 0.5966 | 0.7907 | 0.4521 | 1.3536 |
| AbDiffuser (uniform prior) | 0.8398 | 0.5937 | 0.5742 | 0.7623 | 0.6705 | 1.8365 | 0.6095 | 0.8825 | 0.4795 | 1.0698 |
| AbDiffuser (no projection) | 11.1431 | 11.0062 | 10.8279 | 13.8692 | 14.4139 | 10.4367 | 11.1709 | 15.7536 | 11.5205 | 11.2404 |
| AbDiffuser (no Cov.) | **0.6302** | **0.4011** | **0.3826** | **0.4946** | **0.5556** | 1.6553 | **0.4169** | **0.5585** | **0.4321** | **0.8310** |
| AbDiffuser | 0.5230 | 0.3109 | 0.2862 | 0.3568 | 0.3917 | 1.5073 | 0.3322 | 0.4036 | 0.3257 | 0.7599 |
| AbDiffuser (side chains) | 0.4962 | 0.3371 | 0.3072 | 0.3415 | 0.3768 | 1.3370 | 0.3637 | 0.3689 | 0.3476 | 0.8173 |

Table 4: Detailed RMSD for generated antibodies based on Paired OAS dataset [61]. The top three results in each column are highlighted as **First**, **Second**, **Third**.

| Model | Full ↓ | Fr ↓ | Fr. H ↓ | CDR H1↓ | CDR H2↓ | CDR H3↓ | Fr. L ↓ | CDR L1↓ | CDR L2↓ | CDR L3↓ |
|---|---|---|---|---|---|---|---|---|---|---|
| MEAN [43] | 0.7792 | 0.3360 | 0.3045 | 0.4569 | 0.3359 | 2.9053 | 0.3645 | 0.4425 | 0.2490 | 0.6862 |
| EGNN (AHo & Cov.) | 9.2180 | 8.7818 | 8.5527 | 12.0018 | 12.5770 | 10.0308 | 8.9396 | 14.2269 | 9.5391 | 10.4077 |
| FA-GNN (AHo & Cov.) | 3.1800 | 3.3761 | 1.8529 | 0.6446 | 0.5223 | 2.0202 | 4.2721 | 0.5633 | 0.5376 | 3.4047 |
| AbDiffuser | **0.3822** | **0.2186** | **0.1669** | **0.3611** | **0.2737** | 1.1699 | **0.2610** | **0.1937** | **0.2006** | **0.6648** |
| AbDiffuser (side chains) | 0.4046 | 0.2686 | 0.2246 | 0.3861 | 0.3115 | **1.1191** | 0.3073 | 0.2242 | 0.2379 | 0.7122 |
| AbDiffuser ($\tau = 0.75$) | 0.3707 | 0.2138 | 0.1615 | 0.3541 | 0.2709 | 1.1210 | 0.2563 | 0.1830 | 0.1946 | 0.6615 |
| AbDiffuser (s.c., $\tau = 0.75$) | 0.3982 | 0.2729 | 0.2277 | 0.3914 | 0.2917 | 1.0624 | 0.3127 | 0.2492 | 0.2548 | 0.7131 |
| AbDiffuser ($\tau = 0.01$) | 0.3345 | 0.2000 | 0.1463 | 0.3389 | 0.2723 | 0.9556 | 0.2430 | 0.1530 | 0.1792 | 0.6582 |
| AbDiffuser (s.c. $\tau = 0.01$) | 0.6795 | 0.6168 | 0.5938 | 0.8161 | 0.7113 | 1.1550 | 0.6396 | 0.7938 | 0.7048 | 0.8395 |

Table 5: Detailed RMSD for generated antibodies based on Trastuzumab mutant dataset [55]. The top three results in each column are highlighted as **First**, **Second**, **Third**.

Here, in Tables 4 and 5 we provide per-region distances between folded and optimized structures of the generated sequences and the generated structures. As you can see these distances correlate well with the overall RMSD reported in the main text. As the folded structures are optimized, we also investigate how the error changes if similarly to folding models we also use Rosetta [2] to optimize the generated structures in Tables 6 and 7. As can be seen, the optimization slightly reduces the RMSDs, but the relatively modest change for AbDiffuser-generated structures hints at the fact that

the generated structures were relatively physically plausible. In Table 5 we see that while the model that generates the side chains achieves a worse overall RMSD, in most cases it models CDR H3 more precisely, which is the most important part function-wise. This seeming focus on CDR H3 might explain why this model achieved a better predicted binding probability, even while modeling the overall structure slightly worse.

| Model | Full↓ | Fr↓ | Fr. H↓ | CDR H1↓ | CDR H2↓ | CDR H3↓ | Fr. L↓ | CDR L1↓ | CDR L2↓ | CDR L3↓ |
|---|---|---|---|---|---|---|---|---|---|---|
| EGNN | 9.8129 | 9.3487 | 9.2647 | 13.2206 | 13.1699 | 10.4327 | 9.3722 | 14.8368 | 10.1526 | 10.6565 |
| EGNN (AHo) | 10.1364 | 9.5233 | 9.3961 | 13.3611 | 13.7014 | 12.4793 | 9.5918 | 15.6919 | 10.8115 | 11.3710 |
| EGNN (AHo & Cov.) | 9.5411 | 8.8202 | 8.6761 | 13.0186 | 13.3938 | 12.1843 | 8.8849 | 15.4368 | 10.1352 | 11.2356 |
| FA-GNN | 0.7817 | 0.4844 | 0.4228 | 0.5558 | 0.6293 | 2.1533 | 0.5205 | 0.7222 | 0.4617 | 1.0457 |
| FA-GNN (AHo) | 0.7767 | 0.4116 | 0.3918 | 0.6002 | 0.7031 | 2.2311 | 0.4228 | 0.8372 | 0.4054 | 1.0333 |
| FA-GNN (AHo & Cov.) | 0.7798 | 0.5061 | 0.4541 | 0.5485 | 0.5949 | 1.9846 | 0.5195 | 0.7121 | 0.3914 | 1.2032 |
| AbDiffuser (uniform prior) | 0.8122 | 0.5528 | 0.5300 | 0.7053 | 0.6184 | 1.8129 | 0.5704 | 0.8326 | 0.3914 | 1.0416 |
| AbDiffuser (no projection layer) | 10.9194 | 10.7255 | 10.5253 | 13.6499 | 15.0346 | 10.9846 | 10.9083 | 15.9310 | 11.6059 | 11.7446 |
| AbDiffuser (no Cov.) | **0.5867** | **0.3425** | **0.3206** | **0.4272** | **0.4848** | **1.6261** | **0.3606** | **0.4921** | **0.3296** | **0.7801** |
| AbDiffuser | **0.5068** | **0.2896** | **0.2642** | **0.3282** | **0.3708** | **1.4921** | **0.3110** | **0.3871** | **0.2611** | **0.7334** |
| AbDiffuser (side chains) | **0.4463** | **0.2751** | **0.2426** | **0.2764** | **0.3266** | **1.2869** | **0.3025** | **0.3187** | **0.2390** | **0.7533** |

Table 6: Detailed RMSD for generated antibodies based on Paired OAS dataset [61] after optimization with Rosetta [2]. The top three results in each column are highlighted as **First**, **Second**, **Third**.

| Model | Full↓ | Fr↓ | Fr. H↓ | CDR H1↓ | CDR H2↓ | CDR H3↓ | Fr. L↓ | CDR L1↓ | CDR L2↓ | CDR L3↓ |
|---|---|---|---|---|---|---|---|---|---|---|
| MEAN [43] | 0.7412 | 0.3004 | 0.2718 | 0.5395 | 0.2909 | 2.7830 | 0.3261 | 0.3758 | 0.2593 | 0.6849 |
| EGNN (AHo & Cov.) | 9.2535 | 8.8170 | 8.5701 | 12.0330 | 12.7993 | 10.1256 | 8.9911 | 14.4588 | 9.7059 | 10.6565 |
| FA-GNN (AHo & Cov.) | 2.1631 | 2.2522 | 1.1541 | 0.6734 | 0.5783 | 2.0892 | 2.9101 | 1.4517 | 0.5797 | 2.1591 |
| AbDiffuser | **0.3692** | **0.2017** | **0.1415** | **0.3349** | **0.2474** | 1.1464 | **0.2479** | **0.1743** | **0.1589** | **0.6625** |
| AbDiffuser (side chains) | 0.4087 | 0.2755 | 0.2304 | 0.3632 | 0.3044 | 1.1065 | 0.3141 | 0.2957 | 0.1920 | 0.7217 |
| AbDiffuser ($\tau = 0.75$) | **0.3584** | **0.1969** | **0.1358** | **0.3283** | **0.2459** | 1.1003 | **0.2434** | **0.1642** | **0.1513** | **0.6599** |
| AbDiffuser (s.c., $\tau = 0.75$) | 0.3981 | 0.2747 | 0.2267 | 0.3615 | 0.2795 | **1.0497** | 0.3155 | 0.2939 | 0.2050 | 0.7151 |
| AbDiffuser ($\tau = 0.01$) | **0.3210** | **0.1837** | **0.1202** | **0.3023** | **0.2464** | **0.9288** | **0.2306** | **0.1257** | **0.1285** | **0.6583** |
| AbDiffuser (s.c. $\tau = 0.01$) | 0.6298 | 0.5661 | 0.5450 | 0.7135 | 0.6025 | **1.0870** | 0.5868 | 0.6878 | 0.6012 | 0.8720 |

Table 7: Detailed RMSD for generated antibodies based on Trastuzumab mutant dataset [55] after optimization with Rosetta [2]. The top three results in each column are highlighted as **First**, **Second**, **Third**.

## L  Wet-Lab Validation of Designs

In our wet-lab experiments, we follow the procedure of Hsiao et al. [29]. As we focus on Trastuzumab mutants, the experiments need to account for the fact that the part of the HER2 antigen to which Trastuzumab binds is prone to misfolding in vitro. In the usual experimental procedure for the SPR measurements of Kd values, the antibody is immobilized and the antigen (in this case HER2) is used as a ligand. However, for the domain to which Trastuzumab is binding in HER2, taking into account the problem of misfolding, the standard procedure described by Hsiao et al. [29] is to immobilize the antigen and treat the antibody as a ligand, making the 'misfolded' HER2 essentially invisible in the binding kinetics.

As pointed out by Mason et al. [55], mutating Trastuzumab such that the resulting antibody still binds to HER2 is quite challenging, as some binders in the dataset are just one edit away from non-binders. We also check the similarity of our new binders verified in the wet-lab experiments to both the binder and non-binder sets from the dataset. As can be seen in Table 8 some of the best binders we discovered are equidistant to both sets.

To further highlight the similarity of the non-binder and the binder sets from Mason et al. [55] we include the residue frequency (logo) plots for the CDR H3 positions that were varied in Figure 5.

## M  Implementation and Training Details

As mentioned in the main text, all of the models are evaluated using the same denoising procedure. They also share the input embedding procedure: we utilize a learned dictionary for each amino acid type and another learned dictionary for the chain type (light or heavy). These initial embeddings are summed together to form the input to the actual model. For all models except the APMixer, an

| Generated binder | Binder dist. | Num. closest binders | Non-binder dist. | Num. closest non-binders | KD↓ |
|---|---|---|---|---|---|
| SRYGSSGFYQFTY | 2 | 2 | 2 | 2 | 5.35e-08 |
| SRWLASGFYTFAY | 1 | 1 | 2 | 2 | 4.72e-09 |
| SRWSGDGFYQFDY | 1 | 1 | 2 | 3 | 4.12e-09 |
| SRWRGSGFYEFDY | 1 | 1 | 2 | 3 | 2.89e-09 |
| SRWRASGFYAYDY | 1 | 2 | 3 | 19 | 2.00e-09 |
| SRYGGFGFYQFDY | 2 | 3 | 2 | 2 | 1.86e-09 |
| SRYGGSGFYTFDY | 2 | 8 | 2 | 2 | 3.17e-10 |

Table 8: Edit distances of our generated binders to closest binders and non-binders in the dataset by Mason et al. [55], together with the number of such closest (non-)binders.

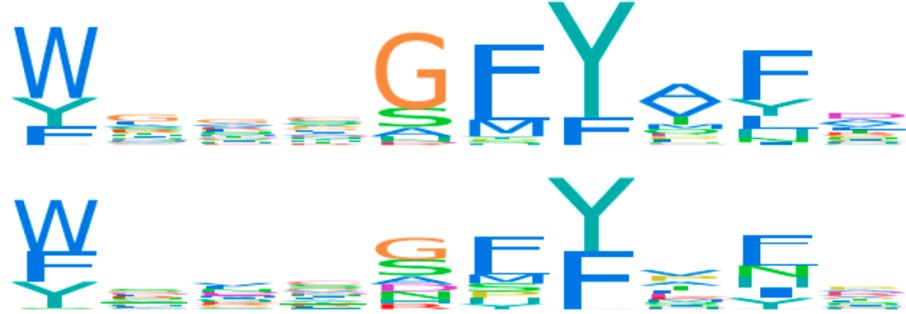

Figure 5: Logo plots for relative amino acid frequencies in the mutated CDR H3 positions for binders (top) and non-binders (bottom) in the dataset by Mason et al. [55].

additional sinusoidal sequence position embedding is included in the sum. The model also receives the time step $t$ as an additional input, concatenated with the other inputs. Similar to Hoogeboom et al. [28], we found this simple time-step conditioning to work well. The input and output of the model are always processed with our constraint layer from Section 3.4. The amino acid type predictions are made by computing a vector product with the input amino acid dictionary.

We will cover the model-specific architectural details in the following.

## M.1 APMixer Details

We build ab mixer out of blocks as described in Section 3.3, where each block consists of two MLPs, that are applied consecutively on the columns

$$X_{\cdot,j} = X_{\cdot,j} + W_2\rho(W_1\text{LayerNorm}(X_{\cdot,j}))\text{for all } j \in [c]$$

and on the rows

$$X_{i,\cdot} = X_{i,\cdot} + W_4\rho(W_3\text{LayerNorm}(X_{i,\cdot}))\text{for all } i \in [r].$$

The whole APMixer used 8 such blocks plus an embedding MLP and two readout MLPs, one for predicting residue atom positions and another for the residue type predictions. Their outputs are respectively processed using equivariant and invariant frame averaging formulations.

As we noted earlier, each block is wrapped in frame averaging. We choose to process half of the output equivalently and half invariantly. It is possible to just use the equivariant formulation throughout the inner blocks, but we found this to slightly improve the performance of the model. Possibly because this aligns slightly better with having position-invariant sequence features. In this case, we extend the APMixer block to have a linear layer before the column MLP, to merge the two representations.

We use 2 layer MLPs throughout, with the embedding dimension of $h = 1920$. The column MLP is applied on three columns at a time to ensure easier modeling of 3D interactions. Also, to facilitate easier modeling of multiplicative interactions (e.g. forces) these MLPs use gated activation in the hidden layer $x = x_{0:h} * \sigma(x_{h:2h})$, where x is the output of the linear layer, $h$ is the embedding dimension (the MLPs hidden dimension is effectively $2 \times h$) and $\sigma$ is the activation function. We chose to use the SiLU [25] activation function as it appears to perform the best on geometric tasks, both on APMixer and the baseline GNN architectures.

### M.2 Baselines

#### M.2.1 Transformer

For the sequence-only generation, we use a standard transformer encoder[12] as implemented by PyTorch [65], with 3 blocks, embedding dimension of 2048, and pre-normalization. The transformer uses the GELU [25] activation function as standard.

#### M.2.2 EGNN

We use the EGNN [75] as implemented for the equivariant molecule diffusion by Hoogeboom et al. [28]. We extend it to work with residues that have many atoms by using the formulation proposed by Huang et al. [31] and also used by Kong et al. [43], which for updating each node (residue) make use of the distances $Z_{ij}^l = X_i - X_j$ between atoms in each residue:

$$
\begin{aligned}
m_{ij} &= \phi_e \left( h_i^l, h_j, Z_{ij}^l (Z_{ij}^l)^T \right), \\
m_i &= \sum_j m_{ij}, \\
h_i^{l+1} &= \phi_h(h_i^l, m_i), \\
X_i^{l+1} &= X_i^l + \sum_j \frac{Z_{i,j}^l}{\|Z_{i,j}^l\|_2 + 1} \phi_x \left( h_i^t, h_j^t, Z_{ij}^l (Z_{ij}^l)^T \right),
\end{aligned}
$$

where $X_i$ is a matrix of $a \times 3$ if we have $a$ atoms per residue.

To improve the model stability, we apply a layer normalization on $h^t$ at the beginning of each such block. We wrap every block in EGNN in our constraint layer Section 3.4. The EGNN has 6 such blocks and uses an embedding dimension of 576. As standard, we use SiLU activation and, similarly to AgentNet, we found it slightly beneficial to use a hidden dimension in the MLP of twice the embedding size. The model uses a readout MLP to make the final embedding which is used to predict the amino acid type. This layer is also preceded by layer normalization. An embedding MLP is used for the inputs.

#### M.2.3 FA GNN

We use the same GNN architecture as used by Puny et al. [68]:

$$
\begin{aligned}
m_{i,j} &= \phi_e(h_i^l, h_j) \\
m_i &= \sum_j m_{i,j} \\
h_i^{l+1} &= \phi_h(h_i^l, m_i),
\end{aligned}
$$

which is essentially the same architecture as EGNN, just without the distance-based edge features and explicit position updates. To process the positions, we wrap each of these GNN blocks using frame averaging, and for the very first embedding MLP we supply residue atom positions as inputs together with the other features.

The FA-GNN has 4 of these GNN blocks and uses an embedding dimension of 384. SiLU activation and the hidden MLP dimension of twice the embedding dimension are used. The FA-GNN uses the same setup as APMixer with two readout heads for amino acid type prediction and atom position prediction.

### M.3 Training

We use AdamW for training [52] with a weight decay of 0.01 and with a learning rate of $2 \cdot 10^{-4}$ for all structure models, while the transformer used a learning rate of $1 \cdot 10^{-4}$. We experimented with weight decay of $1e - 12$ and learning rates in the range of $1 \cdot 10^{-3}$ to $1 \cdot 10^{-4}$ for all models, to determine the chosen values. We also normalize the gradient norm to unit length during training. All models use a batch size of 4. This is the maximum batch size that allows training the baseline models on our 80GB GPU. The APMixer due to the better memory complexity allows for batch sizes up to

32, but to keep the setup similar, we also used a batch size of 4 for it. For paired OAS we train the models for 50 epochs, while for HER2 binder generation we train for 600 epochs, for around 1.25M training steps in both cases. The Paired OAS dataset [61] was split into 1000 test samples and 1000 validation samples. Similarly, the HER2 binder dataset [55] was split to have 1000 test samples and 100 validation samples.

We keep track of the exponential moving average of the model weights, for all models, with a coefficient of 0.995, and use these weights at test time. No model selection is performed.

During training, we use idealized atom positions as returned by our projection layer (Section 3.4) as the target. This ensures that the exact target configuration is reachable by the model.

The CDR redesign baselines of RefineGNN [34] and MEAN [43] were trained for the same amount of steps using the default parameters and early stopping.

All experiments were performed on an 80GB A100 GPU.

### M.4  APMixer Classifier

We adapt APMixer for HER2 binder classification (Section 5.2) by including a global readout. This readout is constructed by applying mean pooling over the frames and residues after each APMixer block and then applying a layer normalization and a linear layer. All of the linear layer outputs are summed to obtain the final prediction. This follows a popular graph classification readout used in GNNs [93].

The classifier is trained for 100 epochs using a batch size of 16. No exponential moving average of weights is used, and the model with the best validation accuracy is selected. The full HER2 dataset was split to have 3000 test samples and 1000 validation samples.

## N  Conditional CDR Redesign

While co-crystal structures usually come quite late in the drug discovery process, performing CDR structure and sequence redesign for a given, unseen co-crystal structure is a popular benchmark for deep generative models [53; 43; 44; 34]. Here, we use the data split from Luo et al. [53], derived from the SAbDab database of antibody-antigen structures [13]. This split ensures that all antibodies similar to those of the test set ($> 50\%$ CDR H3 identity) are removed from the training set. In contrast to the experimental setup of Luo et al. [53], we report the results for the more difficult task of redesigning all CDRs at once[2], rather than redesigning a single CDR at a time. Following Luo et al. [53], the results are averaged over 100 samples for each of the 19 test structures. We observed that in this setup the minimization protocol used by Luo et al. [53] increased the RMSD values (Table 9), therefore, we report the results without it.

To adapt AbDiffuser to this conditional case, we build a bipartite graph between the antibody CDR residues and the nearest antigen residues that are at most 10Å away, in terms of $C_\beta - C_\beta$ distance. Then we simply replace the initial MLP embedding layer with a one-layer FA-GNN. The rest of the APMixer architecture remains unchanged. We trained AbDiffuser with a batch size of 8 for 30k steps, with the rest of hyperparameters staying the same as before.

Table 9 shows that in this conditional setup AbDiffuser outperforms the state-of-the-art diffusion baseline (DiffAb) in amino acid recovery (AA) by a wide margin. This good performance could be attributed to the usefulness of AHo numbering, and APMixer being better at modeling sequences than a GNN. Our ablation experiments on paired OAS generation (Table 1) corroborate this hypothesis. Although the CDR RMSD results are generally quite comparable between DiffAb and AbDiffuser, importantly, AbDiffuser performs better in CDR H3, which has the most variability in its structure and can influence the binding the most. Both methods strongly outperform RosettaAntibodyDesign (RAbD) [2], the state-of-the-art computational biology antibody design program, in terms of amino acid recovery.

---

[2]We use the checkpoint for this task provided by the original authors.

| Model | CDR H1 | | CDR H2 | | CDR H3 | | CDR L1 | | CDR L2 | | CDR L3 | |
|---|---|---|---|---|---|---|---|---|---|---|---|---|
| | AA↑ | RMSD↓ | AA↑ | RMSD↓ | AA↑ | RMSD↓ | AA↑ | RMSD↓ | AA↑ | RMSD↓ | AA↑ | RMSD↓ |
| Rosetta (RAbD)* [2] | 22.85% | 2.261 | 25.5% | 1.641 | 22.14% | 2.9 | 34.27% | 1.204 | 26.3% | 1.767 | 20.73% | 1.624 |
| DiffAb (Minimized)* [53] | 65.75% | 1.188 | 49.31% | 1.076 | 26.78% | 3.597 | 55.67% | 1.388 | 59.32% | 1.373 | 46.47% | 1.627 |
| DiffAb (Minimized) [53] | 66.37% | 1.371 | 42.82% | 1.337 | 28.27% | 3.798 | 62.91% | 1.520 | 62.59% | 1.653 | 49.38% | 1.616 |
| DiffAb [53] | 66.37% | 0.802 | 42.82% | 0.722 | 28.27% | 3.550 | 62.91% | 1.120 | 62.59% | 1.025 | 49.38% | 1.066 |
| AbDiffuser | 79.09% | 1.120 | 72.33% | 0.995 | 36.14% | 2.921 | 85.08% | 1.138 | 85.02% | 1.273 | 75.68% | 0.990 |
| AbDiffuser (side chains) | 76.30% | 1.584 | 65.72% | 1.449 | 34.10% | 3.346 | 81.44% | 1.462 | 83.22% | 1.397 | 73.15% | 1.591 |
| AbDiffuser ($\tau = 0.75$) | 79.76% | 1.083 | 72.96% | 0.950 | 36.53% | 2.805 | 85.60% | 1.098 | 85.71% | 1.237 | 76.29% | 0.955 |
| AbDiffuser (s.c., $\tau = 0.75$) | 76.06% | 1.713 | 66.38% | 1.512 | 34.71% | 3.214 | 81.93% | 1.373 | 83.54% | 1.351 | 73.32% | 1.509 |
| AbDiffuser ($\tau = 0.01$) | 81.11% | 1.075 | 74.27% | 0.946 | 37.27% | 2.795 | 86.26% | 1.115 | 86.85% | 1.238 | 77.06% | 0.966 |
| AbDiffuser (s.c. $\tau = 0.01$) | 75.36% | 2.463 | 66.89% | 2.010 | 35.56% | 3.124 | 83.10% | 1.525 | 82.95% | 1.623 | 74.19% | 1.502 |

Table 9: Generating CDR structure and sequence for existing co-crystal structures from SAbDab [13]. All CDRs are generated at once. The amino acid recovery rate (AA) and RMSD are reported for each individual region. Baseline models from Luo et al. [53] marked by * generated CDRs one at a time.

