# OpenReview forum: "AbDiffuser: full-atom generation of in-vitro functioning antibodies"
_NeurIPS.cc/2023/Conference — NeurIPS 2023 spotlight_

### Official Review · Reviewer_NpkL · 2023-07-01

**Soundness:** 3 good
**Presentation:** 2 fair
**Contribution:** 2 fair
**Rating:** 6
**Confidence:** 2

**Summary:**

This paper introduces AbDiffuser, a new equivariant diffusion model, designed to generate full antibody structures efficiently and with high quality. By incorporating family-specific priors and utilizing a novel neural network architecture called APMixer, AbDiffuser demonstrates improved performance in modeling the sequence and structural properties of natural antibodies. Experimental validation shows that the generated antibody binders have high affinity, improving the affinity of a cancer drug. These findings highlight the potential of generative models in creating high-affinity antibody binders without the need for post-selection by binding predictors.

**Strengths:**

* The proposed approach outperforms other baseline methods across various metrics related to antibody designing. This could indicate potential effectiveness in generating high-quality antibody structures.

* The novel architecture introduced in this paper exhibits a lower memory footprint during training, which is a significant advantage compared to the baseline architectures. This not only reduces resource requirements but also enhances the efficiency of sample generation, making it a more practical and scalable solution.

* The design choices made in this work are backed by theoretical results and are further supported by in silico experiments. The alignment between theoretical motivations and experimental outcomes adds credibility to the overall approach.

**Weaknesses:**

* The core components of the proposed model, including the diffusion model and the mixer model, bear a strong resemblance to corresponding reference models. While there are some minor additions, such as the physics-informed residue representation, AHo-specific residue frequencies, and encoding conditional atom dependencies, these modifications may be viewed as relatively insignificant. This could raise questions about the extent of novelty and originality introduced by the proposed model.

* In lines 138-139, it would be beneficial for the authors to provide further insights or explanations regarding why the second choice is expected to improve the performance. This would enhance the clarity and understanding of the motivations behind the model design decisions.

* A minor typographical error is present at line 173, where a missing space between "\AA" and "while" is observed. While this is a minor issue, attention to detail and thorough proofreading are important aspects of a scientific publication.

**Questions:**

Table 2 demonstrates that having more parameters is beneficial, contrary to the general notion that fewer parameters are desirable. It raises the question of why an increased number of parameters is advantageous in this specific context and how it leads to improved performance. Could the authors elaborate on the reasons behind the observed benefit of having a larger parameter count in their model?

**Limitations:**

The choice of HER2 binder design presented in the paper could potentially be limited to being just one successful case study. It is important to acknowledge that the success achieved in this particular case study does not guarantee similar outcomes in other design scenarios. Providing a clear remark about this limitation would be valuable for the scientific community, considering that the drug discovery process is known to be expensive and challenging. It would help manage expectations and promote a realistic understanding of the potential generalizability and applicability of the proposed methodology beyond the specific case study presented.

---

> ### Author Rebuttal · Authors · 2023-08-10
>
> Thank you for your thoughtful review. We address your comments in turn.
>
> > The core components of the proposed model, including the diffusion model and the mixer model, bear a strong resemblance to corresponding reference models. While there are some minor additions, such as the physics-informed residue representation, AHo-specific residue frequencies, and encoding conditional atom dependencies, these modifications may be viewed as relatively insignificant. This could raise questions about the extent of novelty and originality introduced by the proposed model.
>
> From a neural network architecture perspective, our main contribution lies in the “table” representation we propose and not in the specific combinations of layers we employ. Our approach stands in contrast to current relational approaches (GNNs and transformers), is proven to be universal (whereas for usual relational models such as equivariant GNNs universality is not guaranteed (https://arxiv.org/pdf/2301.09308.pdf), and yields demonstrable practical benefits (1-order of magnitude less memory, improved generative model quality, seamless handling of sequence length changes).
>
> At the same time, it is crucial to stress that our work goes beyond the proposal of a neural network and innovates both in the generation aspect (with the introduction of the informative priors and the proof of the benefit they bring in Theorem E.1) and the physics informed residue representation (with the projection layer). The former is shown to carry strong empirical benefits, and the latter makes it for the first time possible to respect the known geometry constraints of protein residues while operating in an external coordinate frame (i.e., coordinates and not angles), thus allowing us to use Gaussian diffusion while respecting the constraints and avoiding the more inaccurate IGSO3 diffusion (https://openreview.net/pdf?id=BY88eBbkpe5).
>
> In our view, the current antibody generative models are more direct adaptations of graph or sequence generative models, while in this work we aimed to design an approach where each component was specifically made with antibodies in mind.
>
> > In lines 138-139, it would be beneficial for the authors to provide further insights or explanations regarding why the second choice is expected to improve the performance. This would enhance the clarity and understanding of the motivations behind the model design decisions.
>
> The main reason for applying the frame averaging on every layer separately was that it performed better in our experiments. We don’t have a mathematical reason to explain this.
>
> > A minor typographical error is present at line 173, where a missing space between "\AA" and "while" is observed. While this is a minor issue, attention to detail and thorough proofreading are important aspects of a scientific publication.
>
> Thank you for pointing this out. We will make sure to proofread the paper a few more times for the final version.
>
> > Table 2 demonstrates that having more parameters is beneficial, contrary to the general notion that fewer parameters are desirable. It raises the question of why an increased number of parameters is advantageous in this specific context and how it leads to improved performance. Could the authors elaborate on the reasons behind the observed benefit of having a larger parameter count in their model?
>
> As far as we are aware, the good performance of deep neural networks is attributed to overparameterization, as it is presumed to smooth the loss landscape (e.g. see http://proceedings.mlr.press/v119/buhai20a/buhai20a.pdf).
> In addition to this, the biggest risk of having too many parameters is higher potential of overfitting but this is less of an issue for generative/diffusion models.
> It is also the case that, for example, large language generative model performance increases with size (e.g. Table 3 in https://arxiv.org/pdf/2307.09288.pdf).
>
> > The choice of HER2 binder design presented in the paper could potentially be limited to being just one successful case study. It is important to acknowledge that the success achieved in this particular case study does not guarantee similar outcomes in other design scenarios. Providing a clear remark about this limitation would be valuable for the scientific community, considering that the drug discovery process is known to be expensive and challenging. It would help manage expectations and promote a realistic understanding of the potential generalizability and applicability of the proposed methodology beyond the specific case study presented.
>
> It is true that drug design is a complicated endeavor, that there are many variables that can impact the outcome, and that success is not guaranteed. Our model, like any other, is not exempt from this: naturally, the achieved binding rates and the required dataset sizes could differ quite a bit depending on the problem at hand. We believed that this was already clear, but we will be happy to make it explicit.
>
> The fact that our model does well on general antibody generation (paired OAS) gives some hope that it is able to properly capture many different aspects of antibodies and that the results would transfer to some extent.
>
> To also evaluate our model in different scenarios, we tested our model on antigen conditioned CDR inpainting on the SAbDab split used in DiffAb. You can find the results in the general response. It can be seen that our model does quite well on this task, which further increases our hope of it generalizing to other antigens and problems.

---

> > ### Comment · Reviewer_NpkL · 2023-08-18
> > **Thanks for the clarification**
> >
> > I extend my gratitude to the authors for their comprehensive response that effectively addresses my previous concerns. The additional details provided have clarified the points of contention, ensuring a more robust understanding of the manuscript. Consequently, I find it appropriate to raise the score for this submission.

---

### Official Review · Reviewer_9R9d · 2023-07-01

**Soundness:** 2 fair
**Presentation:** 1 poor
**Contribution:** 3 good
**Rating:** 5
**Confidence:** 4

**Summary:**

The paper proposes AbDiffuser, a new diffusion model for antibody structure and sequence generation. AbDiffuser proposes several ideas to improve generation: (i) Euclidean diffusion with frame averaging to achieve SE(3) equivariant diffusion, (ii) antibody sequence alignment to standardize sequence length across antibodies, (iii) atom to rigid body projections to maintain inter-residue physical constraints, and (iv) informative priors when sampling sequence and structure. The method is evaluated on the Observable Antibody Space and HER2 binder dataset against a selection of prior works but not state-of-the-art baselines. Performance is measured with how close the generated sequences match statistics of the test set as well as some physics based metrics. The authors ablate most components of the method to find it improves performance. Lastly, they demonstrate 16 of their designs fold into antibodies and have high binding affinity to HER2.

**Strengths:**

- AbDiffuser explores many ideas that are different to current state-of-the-art protein generation diffusion models. (1) Frame averaging for conferring SE(3) equivariance to MLPMixer over positions. (2) Aligning antibodies to a reference sequence for handling variable lengths. (3) Projecting coordinates to rigid body constraints on input and output of the denoising model. (4) Informative priors over the coordinates and sequence based on training set statistics. Each component of the method seems to help based on the task and metrics defined in the paper.

- I appreciate the wet-lab validation of their designs which is almost never present in protein papers at machine learning conferences.





**Weaknesses:**

- **Biggest issue: poor presentation**. There are simply too many new ideas that comprise AbDiffuser and all the details are left to the appendix. The reader should not have to go through 28 pages (main text and appendix) to understand the method. Even after looking through the appendix, I am unsure what the full algorithm is. In fact, there is no algorithm or figure of how AbDiffuser works from training to sampling.

- **Missing motivation for frame averaging**. Frame averaging (FA) is introduced without motivation. On line 37-38, the authors state previous methods that confer SE(3) equivariance and physical constraints come with "increased complexity". These complexities are never spelled out. How is AbDiffuser simpler? The paper proposes frame averaging with informative priors *and* the rigid body projection step. This seems quite complicated to me. Additional points on FA:

- **Missing related work**.
  - FA for proteins. the authors fail to cite [1] which already proposed FA for proteins.
  - The projection step seems related to the projection step, "Equivariant Consensus Structure from Weighted Inter-residue Geometries", in Appendix L of Chroma but this connection is not mentioned.

- **Gaussian diffusion plus projection step vs. SE(3) diffusion**. The projection step seems to me as a way to circumvent working with rigid body frames by always projecting the gaussian diffusion over coordinates onto the rigid body constraints. But this seems close to SE(3) diffusion [2] over frames where the C-a coordinates diffuse along with the backbone atoms which always have the rigid body constraints. Why is the gaussian diffusion and projection step necessary to develop if we have SE(3) diffusion? Furthermore, SE(3) diffusion is not compared with so we do not know which is better.

- **Unjustified claims**. I have concerns with several claims in the paper.
  - Line 3 claims AbDiffuser uses a new representation of protein structure which as I understand is the FA component. But as stated this has already been done [1].
  - Line 5-6 "Our approach improves protein diffusion ... handles sequence-length changes". This is not true. The authors achieve this by aligning all antibodies to a reference alignment (the AHo numbering) that has been specifically developed for antibodies. This does not exist for nearly all proteins so their method cannot be applied. Even if it can be applied, the authors cannot claim it works for other proteins until it is demonstrated.
  - Line 67 - 69 "These results provide the first experimental evidence that a generative model trained on mutagenesis data can create new antibody binders...". This is not true. [4] is just one paper that showed new antibody binders can be created with generative models. In addition, RFdiffusion [5] which can generate de novo binders is not mentioned.
  - Title "AbDiffuser: full-atom generation of in-vitro functioning antibodies". This a bold claim that is **not justified** by the results of generating 16 functioning antibodies for a single target HER2. Furthermore, it is unclear how novel these antibodies are.
  - Line 87. The authors claim APMixer is a novel neural network for processing proteins. The claimed novelty seems very incremental. MLPMixer is not new, FA is not new. The only component which I believe is novel is the sequence alignment of each antibody such that they all have the same sequence length as input. But this does not warrant technical novelty. Furthermore, I don't understand why SE(3) universality isn't already provided by FA.

- **Unclear experimental set-up**. The experimental set-up is poorly written. The metrics are not clearly defined in the main text. The training dataset is unclear: OAS does not have structures so how are the structures obtained for training? Do the authors train RefineGNN and MEAN baselines only on the CDR which they were designed for or also give them the full sequence? How are the splits performed? The Wasserstein distance metric has no equation so I am unclear what it is exactly measuring. Why is there no metric on diversity and novelty of the sequences? Perhaps this information is buried in the appendix but it should be up front in the main text.

- Line 250-252. Why would we expect IgLM to have the best accuracy just because it was trained on more data? IgLM is not trained on the structure like AbDiffuser or the other baselines.

- **Experimental results**. The authors claim they have designed binders which have been tested in the wet-lab but this result must be accompanied by experimental results in the supplementary or plots in the supplement. This is standard in bio journals. In addition it is unclear how novel each design is. If only a few mutations were necessary to obtain a tight binder then this task is not difficult.

- **Lack of baselines**. The authors miss out on state-of-the-art. On top of the ones already discussed there have been diffusion models for antibody generation [6, 7].

[1] https://arxiv.org/pdf/2301.10814.pdf

[2] https://arxiv.org/abs/2302.02277

[3] https://www.biorxiv.org/content/10.1101/2022.12.01.518682v1.full.pdf

[4] https://www.biorxiv.org/content/biorxiv/early/2023/03/29/2023.01.08.523187.full.pdf

[5] https://www.biorxiv.org/content/10.1101/2022.12.09.519842v1

[6] https://www.biorxiv.org/content/10.1101/2022.07.10.499510v2

[7] https://arxiv.org/abs/2302.00203

**Questions:**

My questions have been written above.

**Limitations:**

Limitations are not discussed though I believe there are many issues with the paper.

---

> ### Author Rebuttal · Authors · 2023-08-10
>
> Thank you for your in depth questions. We urge you to carefully consider our reply as it addresses the issues you raised and used to motivate your low score.
>
> > Biggest issue: poor presentation. There are simply too many new ideas that comprise AbDiffuser and all the details are left to the appendix. The reader should not have to go through 28 pages (main text and appendix) to understand the method. Even after looking through the appendix, I am unsure what the full algorithm is. In fact, there is no algorithm or figure of how AbDiffuser works from training to sampling.
>
> Thank you for appreciating the number of contributions the paper makes. Due to the limited space we naturally focus on our main contributions in the main paper and delegate less innovative parts, such as the definition of the metrics and the analysis of the results to the appendix. To improve the at-a-glance understanding of the method in the PDF attached to the main response we included sampling and training pseudo algorithms. We will also include them in the final version.
>
> > Frame averaging (FA) is introduced without motivation.
>
> The motivation for frame averaging is twofold 1) the original paper has shown it to compare favorably to alternatives 2) it allows us to conveniently deal with SE(3) equivariance and frees us to utilize a non-relational equivariant model using well understood an easy-to-train blocks (MLPs).
>
> > On line 37-38, the authors state previous methods that confer SE(3) equivariance and physical constraints come with "increased complexity". These complexities are never spelled out. How is AbDiffuser simpler? The paper proposes frame averaging with informative priors and the rigid body projection step. This seems quite complicated to me.
>
> We refer to the fact that building neural networks that work on rotation manifolds is complicated by the non-euclidean structure of the manifold. There are many options for how the rotations can be represented and what losses can be used. These choices have big implications on the model performance, and usually rely on approximations. We elaborate on these approximations for the SO(3) diffusion in our answer to one of your follow up comments.
>
> Another point is that SE3-equivariant GNNs are limited in their ability to capture global geometry (https://arxiv.org/pdf/2301.09308.pdf). Further, these GNN and transformer-based models are well known to require a lot of memory which hinders them from scaling to the full-atom generation we do here. We also witnessed the expressivity issues of GNNs in our numerical experiments, where the GNN models consistently achieved worse RMSD than AbDiffuser variants.
>
> Admittedly, “complexity” was too general a term here. In the final version we will write that “ .. they can require various approximations and architectural considerations that tend to lead to an increased memory complexity.”.
>
> On the other hand, informative priors do not have anything to do with the model complexity or the complexity of the overall approach. They are add-ons that could potentially be added to any diffusion model. We just show (theoretically and practically) that it makes sense to add this and how to build such data-driven priors. As we show in Table 1, models with such priors removed still function well, albeit with reduced performance.
>
> > FA for proteins. the authors fail to cite [1] which already proposed FA for proteins.
>
> Thank you for the reference, we will include it. Note that the authors of this work apply FA to the usual relational self-attention model. We use FA to build a non-relational model, which, as shown in Table 2, has significantly smaller memory/time consumption.
>
> > The projection step seems related to the projection step, "Equivariant Consensus Structure from Weighted Inter-residue Geometries", in Appendix L of Chroma but this connection is not mentioned.
>
> As the title of the referenced chapter suggests, the paper you referenced concerns interactions between residues. This is different from our projection step, which handles the constraints within each residue. Specifically, their approach changes the model to output a specification for an optimization problem that is then solved to build the protein structure (similar to how AlphaFold1 outputs a distance matrix that is later used to find atom positions by gradient descent). In our case the model directly outputs the large-scale protein structure, similarly to for example AlphaFold2. So we don’t really see any similarity between our projection layer and the chapter you reference: the two approaches solve different problems in different ways.
>
> > Line 3 claims AbDiffuser uses a new representation of protein structure which as I understand is the FA component. But as stated this has already been done [1].
>
> The representation we propose goes beyond frame averaging (which can also be employed with GNN & transformers as we and others have demonstrated).
>
> We are proposing to represent members of an aligned protein family by a fixed size “table”, which holds amino acid types and all atom positions and can accommodate any antibody and any side-chains. To achieve this, we rely on a global family alignment scheme and the 4-atom sidechain representation we introduce in Figure 1 and lines 176-200. A key distinction of our representation is that it goes against the common practice of thinking of residues as a set (attention / GNNs) or a variable-length sequence (when adding positional encodings). As we explain in the paper, our representation simplifies learning by allowing the model to associate specific structural roles to particular table rows as well as to easily change the length of a protein mid-generation. The practical advantages of our representation are consistently illustrated in our experiments.
>
> As you have raised a lot of questions there is not enough space in the rebuttal to answer them all. We will answer the remaining ones in the comments bellow, as soon as they are enabled.

---

> > ### Author Response · Authors · 2023-08-10
> > **Additional Answers**
> >
> > > Gaussian diffusion plus projection step vs. SE(3) diffusion. The projection step seems to me as a way to circumvent working with rigid body frames by always projecting the gaussian diffusion over coordinates onto the rigid body constraints. But this seems close to SE(3) diffusion [2] over frames where the C-a coordinates diffuse along with the backbone atoms which always have the rigid body constraints. Why is the gaussian diffusion and projection step necessary to develop if we have SE(3) diffusion? Furthermore, SE(3) diffusion is not compared with so we do not know which is better.
> >
> > Indeed, the projection layer provides a way to avoid using SE(3) diffusion over the frame rotation. We want this as diffusion on SE(3)/SO(3) is imprecise (see below), which contrasts with the usual Gaussian diffusion we are using.
> >
> > Note that technically the diffusion should be SO(3) not SE(3) as chirality of amino acids is fixed in the body (reflections are not allowed).
> >
> > The proper distribution for diffusion on SO(3) is the IGSO(3) (https://openreview.net/pdf?id=BY88eBbkpe5). Unfortunately, the latter does not come with an analytical expression for the density, which is expressed as an infinite series. Thus, to employ diffusion on IGSO(3) one is practically forced to truncate the infinite series rendering the computation cumbersome and imprecise (RFDiffusion retains 2000 terms of that infinite series at every step hinting to a non-convergent series). Note further that RFDiffusion uses a MSE training loss that, by their admittance, is not the mathematically correct choice for the reverse process — possibly because MSE, which is the correct loss for Gaussian diffusion, works better. Also, their formulation depends on an approximation of q(r^(0) | r^(t)) by a point mass to recover an approximate gradient on IGSO(3). So SO3 diffusion in practice comes with multiple approximations and inaccuracies.
> >
> > DiffAb uses SO(3) diffusion and MEAN also uses the Ca + rotation representation of the frame. As can be seen in Table 3 in the paper and the main response here, we do outperform those approaches.
> >
> > > Line 5-6 "Our approach improves protein diffusion ... handles sequence-length changes". This is not true. The authors achieve this by aligning all antibodies to a reference alignment (the AHo numbering) that has been specifically developed for antibodies. This does not exist for nearly all proteins so their method cannot be applied. Even if it can be applied, the authors cannot claim it works for other proteins until it is demonstrated.
> >
> > Thank you for pointing this out. We meant to say antibodies in this line. We will correct it.
> >
> > As mentioned in the conclusions, although we focused here on antibodies, there is no fundamental obstacle for our method to be applied to any sufficiently large protein domain family, as organized in CATH and Pfam. Some of the more ubiquitous enzyme families have similar folds and catalytic functions but substantial substrate diversity (e.g., SAM-dependent methyltransferases, long chain alcohol oxidases, zinc-dependent alcohol dehydrogenases, amine dehydrogenases, and many more enzyme families each have several hundred thousand to millions of sequences in their annotated family each). The TIM-barrel fold, a fold with vast biosynthetic potential, has >20 distinct families in CATH, most sharing stereotypical anatomy with 8 parallel strand alpha-beta (n=8, S=8) fold; our model could be also applied to this family. We intend to explore these possibilities in future work.
> >
> > > Line 67 - 69 "These results provide the first experimental evidence that a generative model trained on mutagenesis data can create new antibody binders...". This is not true. [4] is just one paper that showed new antibody binders can be created with generative models. In addition, RFdiffusion [5] which can generate de novo binders is not mentioned.
> >
> >
> > Our precise claim is more nuanced than what was depicted in the comment:
> > “These results provide the first experimental evidence that a generative model trained on mutagenesis data can create new antibody binders of high affinity without post-selection by learned or physics-based binding predictors”
> >
> > Paper [4] provides very little information about how the generation works but admits to filtering based on an ensemble of models. Other papers have already achieved functional abs with sequence based methods and using filtering (e.g., Mason et al, DOI: 10.1038/s41551-021-00699-9). RFDiffusion has designed binders that were verified in the lab, but they designed ones based on helical bundles and not antibodies. We will elaborate on this in the manuscript.

---

> > ### Author Response · Authors · 2023-08-10
> > **Additional Answers**
> >
> > > Title "AbDiffuser: full-atom generation of in-vitro functioning antibodies". This a bold claim that is not justified by the results of generating 16 functioning antibodies for a single target HER2. Furthermore, it is unclear how novel these antibodies are.
> >
> > We would like to draw the attention to that we rely on the gold standard SPR assay which provides high confidence measurements and is orders of magnitudes more expensive than high-throughput experiments. So the binding results we report are reliable and reproducible. The high cost of SPR experiments makes a higher-throughput experimentation prohibitive. Even bio journal papers often only perform SPR on tens of their best designs (e.g., Mason et al, DOI: 10.1038/s41551-021-00699-9 tested 30 sequences this way).
> >
> > All of the binders we found were novel. You can see the corresponding analysis in Appendix K. There, it is shown that the best binder found is equidistant to both the closest binder and the closest non-binder. Finding a better or even a competitive binder to trastuzumab is no trivial task, as a lot of engineering and optimization went into the latter’s development. Please also note that the best binder reported by [4] which also focused on the HER2 target has pKD of 9.03 which is half a log-unit below our best binder that has a pKD of 9.5.
> >
> > > Line 87. The authors claim APMixer is a novel neural network for processing proteins. The claimed novelty seems very incremental. MLPMixer is not new, FA is not new. The only component which I believe is novel is the sequence alignment of each antibody such that they all have the same sequence length as input. But this does not warrant technical novelty.
> >
> > It is true that the model is largely built from known components. However, almost all new models used for proteins (and even in other domains) also rely mostly on existing components with some improvements. In our eyes, the key distinction of APMixer is that it’s not a relational model (GNN or Transformer), which contrasts with existing structure-based antibody and protein models. As we show in our work, there is merit to using such a non-relational model as it offers better computational efficiency (Table 2), better memory complexity  of O(nd + d^2) compared to O(n^2 + nd + d^2) for traditional transformer or O(Δnd + d^2) for GNNs with message functions (here Δ is degree, n number of nodes/residues and d is the embedding dimension), and better results (Tables 1 and 3). APMixer also avoids the potential limitations of expressive power (see our next comment) that as discussed before can be an issue to GNNs. Furthermore, due to the antibody representation we introduced, APMixer does not need to be conditioned on sequence length. Our aim here is to provide the community with a valid alternative to the usual GNN or Transformer models, which so far has not really been explored.
> >
> > Nonetheless, as you have also stated, the paper introduces many new concepts. So the overall novelty of the work is high and transcends the APMixer architecture alone.
> >
> > > I don't understand why SE(3) universality isn't already provided by FA.
> >
> > FA only provides universality if the model it wraps is already universal. Relational models, such as GNNs, are known to have limitations in terms of their expressive power (see e.g., https://arxiv.org/pdf/1810.00826.pdf, https://arxiv.org/pdf/2301.09308.pdf). As we prove in Appendix G, APMixer does not suffer from these limitations and is actually universal. To the best of our knowledge, this was not previously known for MLP-Mixer and it was not discussed in the original paper. As you can also see in Appendix G, the proof is not trivial.
> >
> > > Line 250-252. Why would we expect IgLM to have the best accuracy just because it was trained on more data? IgLM is not trained on the structure like AbDiffuser or the other baselines.
> >
> > We only claim that IgLM should have close to the best accuracy achievable by the *sequence-only* models (line 254), especially because it was trained on more data and on data that include our test set. Note that the transformer we use in our benchmarks is also a sequence-only transformer, but trained in our framework and with the usual training set.
> >
> > > The authors miss out on state-of-the-art. On top of the ones already discussed there have been diffusion models for antibody generation [6, 7] (dyMEAN and DiffAb)
> >
> > The code for dyMEAN was not available at the time of the submission, but it is true that it's a worthy baseline to include (even though dyMEAN is not a diffusion model). We have included results for dyMEAN and DiffAb in the main response which showcase the superiority of our approach.

---

> > ### Author Response · Authors · 2023-08-10
> > **Additional Answers**
> >
> > > Unclear experimental set-up. The experimental set-up is poorly written.
> >
> > >> The metrics are not clearly defined in the main text.
> >
> > You can find a comprehensive description of the metrics in Appendix I. We will expand the description in the main text (lines 256-263) in the final version, subject to space limitations.
> >
> > >> The training dataset is unclear: OAS does not have structures so how are the structures obtained for training?
> >
> > As stated in line 271, they are folded with IgFold and optimized with Rosetta.
> >
> > >> Do the authors train RefineGNN and MEAN baselines only on the CDR which they were designed for or also give them the full sequence?
> >
> > As RefineGNN and MEAN were built to do CDR inpainting, in this case we train them only on CDR H3 redesign. So, at evaluation time, they receive the test antibody structure and sequence and then fill in the removed CDR H3 (its sequence and structure). In contrast, the other models are also tasked with re-designing the framework.
> >
> > >> How are the splits performed?
> >
> > The experiments on OAS and HER2 binders rely on an i.i.d. splits, which is the mathematically correct choice when aiming to test the ability of a generative model to sample from the same distribution. (A perfect generative model should produce samples that are indistinguishable from true hold out i.i.d. samples.) For the SAbDAb experiment (see general reply) where we aim to test generalization, we adopt the DiffAb split which clusters all antibodies in SAbDab with up-to 50% CDR H3 identity, selects 5 clusters containing 19 distinct antibodies. This ensures that the training set does not overlap with the test set, as SAbDab has many repeating or highly similar entries.
> >
> > >> The Wasserstein distance metric has no equation so I am unclear what it is exactly measuring.
> >
> > The Wasserstein distance, also known as Earth Mover's Distance, is a standard metric to compare distributions (https://docs.scipy.org/doc/scipy/reference/generated/scipy.stats.wasserstein_distance.html) and is generally considered as one of the best ways to evaluate generative models. For example, the popular FID score used to evaluate image generators is effectively a Wasserstein distance under an extra Gaussian assumption. We will elaborate on the definition in the appendix.
> >
> > >> Why is there no metric on diversity and novelty of the sequences? Perhaps this information is buried in the appendix but it should be up front in the main text.
> >
> > All the generated sequences were 100% novel. As we state in line 327, MEAN was the only tested model which did not generate perfectly unique sequences. In its case, only 38.9% of the generated sequences were unique. We will highlight that further in the main text.
> >
> > We also measure novelty through the ``closeness’’ metric, which compares the edit distances of the generated sequences to any of the training sequences with the edit distances of the test sequences to any of the training sequences. Put simply, we compute the edit distance histogram (to the training set) for generated data and test data, and then check how close the two histograms are (using the standard Wasserstein distance). Intuitively, when we are aiming to fit some distribution, what our model produces should be indistinguishable from a hold-out i.i.d set: the generated sequences should neither be too close, nor too far from the training data, but the their distance should follow the same statistics as the i.i.d. hold-out set, corresponding to a small closeness value.
> >
> > > The authors claim they have designed binders which have been tested in the wet-lab but this result must be accompanied by experimental results in the supplementary or plots in the supplement. This is standard in bio journals. In addition it is unclear how novel each design is. If only a few mutations were necessary to obtain a tight binder then this task is not difficult.
> >
> > We include the generated binder sequences as well as the experimentally measured binding affinity values (expressed as Kd - dissociation constants) and experimental result statistics in Figures 3 and 2. We also provide additional information in Appendix K, such as the edit distances to closest known binders and non-binders in the HER2 dataset we have used. For example our best discovered binder has CDR H3 edit distance of 2 from both the binder and non-binder sets. So it's not a trivial mutation, because it is easy to ‘ruin’ the binding. Also, none of our generated 1k binder candidates appeared in the training set.
> >
> > We believe this covers what is usually presented in comparable experimental papers in bio journals (see e.g., [4]). At the same time, please consider that you are reviewing for a ML conference not a bio journal, and that no other comparable ML conference paper has presented in vitro experimental results as we do.

---

> > ### Author Response · Authors · 2023-08-10
> > **Additional Answers**
> >
> > > Limitations are not discussed though I believe there are many issues with the paper.
> >
> > As we stated in the paper (e.g., Chapter 3 lines 87-88), the model we introduce can only work on large homologous protein families and that, while there potential are quite a few such families, we have only demonstrated its utility for antibodies. We see this as the main limitation of our work. Our reply provided thorough answers to the issues you have raised.

---

> > > ### Comment · Reviewer_9R9d · 2023-08-16
> > > **Response**
> > >
> > > Thank you for the response and answers to my questions.
> > >
> > > I agree there is a high level of novelty with AbDiffuser. However, each new components requires thorough investigation with baselines for each. The reader needs a takeaway of why each component is necessary both empirically and theoretically (if possible). I initially was not convinced with the reliance on theoretical arguments rather than running head-to-head experiments. I appreciate the additional comparison to DiffAb which allows comparison to SO(3) diffusion and IPA. This addresses my concerns with unexplained novelty. However, it is not clear how meaningful each improvement is over DiffAb (also the naming of DiffAb vs. AbDiffuser is unfortunate).
> > >
> > > It is unfortunate there cannot be manuscript updates to see what the updated presentation of the method is. The pseudocode in the uploaded PDF alleviates understanding the method. Including code for the "Project" step will also be informative. Clarifications to my questions such as how splits were performed are important to include.
> > >
> > > Thank you for the clarification on experimental validation. While this is a ML venue, I believe details on the experimental validation needs to be rigorously provided, even at the level of what journals require, to not draw criticism from the bio community. I missed the first sentence of section K stating you follow the procedure of Hsiao et al (hopefully you understand, there is a lot of understand).
> > >
> > > My low score was reflective of each of these criticisms. The authors have done a lot of work to address them so I have raised my score. I would have liked to see an updated manuscript with the additional justification and results but I understand that is not in the authors' control. In general, the paper is quite dense, to the point that it was difficult to understand each component in given the time frame.

---

### Official Review · Reviewer_is6j · 2023-07-05

**Soundness:** 4 excellent
**Presentation:** 3 good
**Contribution:** 3 good
**Rating:** 7
**Confidence:** 4

**Summary:**

The authors propose a new approach, called AbDiffuser, that improves protein diffusion by leveraging domain knowledge and physics-based constraints. They follow MLPMixer's architecture to reduce memory complexity by an order of magnitude, enabling backbone and side chain generation. They validate AbDiffuser in silico and in-vitro.

**Strengths:**

- APMixer architecture significantly reduces GPU memory, which is an interesting improvement since many related works rely on a triangular update mechanism that has O(N3) complexity. It is interesting whether such an approach can be adopted for more general tasks, such as protein folding and design.
- The projection layer looks interesting, as it helps maintain the intermediate diffusion state on the "manifold" of valid structures. Similar to the above, it remains unknown whether this may benefit general protein design.
- The authors validate the method in-vitro, increasing the solidness and usefulness.

**Weaknesses:**

- AbDiffuser does not directly use epitope information, which may harm generalization. For example, given a newly seen antigen, how can this method be applied to design an antibody that has a high binding affinity?
- Since the sequence dataset is folded with IgFold (line 271), it may indicate that AbDiffuser's "folding" performance is bounded by IgFold. The model is ranked with another classifier trained with binding probability (line 316). It appears that the IgFold and classifier act as the model's "teacher." How does the model outperform its "teacher"? If we have such a "teacher", why not try to use it directly, with MCMC, Bayesian optimization, etc?
- In line 314, a random mutation approach may generate 9k binders and 25k non-binders, indicating that random mutation causes a success rate of 9/(9+25)=26%. In line 66, AbDiffuser has a success rate of 22.2%. Although the number may increase to 57.1% by post processing, does this mean that "AbDiffuser performs worse than a random mutation baseline"?

**Questions:**

- During forward diffusion, is the residue type continuous or discrete? If it is continuous, how do you perform "backbone/side-chain projection" (section 3.4) since the type of residue is undecided? If it is discrete, during the forward process (by interpolating between a prior distribution and the final one-hot distribution), will the residue type remain unchanged? For example, assuming there are 5 amino acid types, the true state is (0, 0, 0, 0, 1) and the prior distribution is (0.2, 0.2, 0.2, 0.2, 0.2), the residue type will always be the 5th state.
- Are the priors computed over the entire dataset, instead of some antibody family? What would be the performance if directly sampling from the sequence and structure prior?

---

> ### Author Rebuttal · Authors · 2023-08-10
>
> Thank you for your insightful comments, we answer them below.
>
> > AbDiffuser does not directly use epitope information, which may harm generalization. For example, given a newly seen antigen, how can this method be applied to design an antibody that has a high binding affinity?
>
> This is a pertinent question. As explained in lines 359-361, to create diversified and improved binders for a new antigen, we would need a starting dataset of potential binders. Thankfully this can be relatively easily acquired via high-throughput experiments (yeast & phage display, immunogenic campaigns). Due to their speed and low-cost, these experiments are the 1st step in drug discovery taken to roughly scope the Ab landscape when looking for a new drug. Starting with this collection of potential binders, the aim of the HER2 experiment is to select diversified high-binding molecules that, down the line, can be tested for good developability properties. On the other hand, co-crystal structure determination (needed by the current CDR-redesign methods such as DiffAb or MEAN) is usually done only for valuable identified leads.
>
> Nevertheless, to prove that AbDiffuser can also bring benefits to the conditional generation task, we performed a new experiment on SAbDab CDR-redesign, where we compare against DiffAb and Rosetta (see general reply). The method extension was achieved by adding a GNN that performed information exchange between Ag and Ab as was done previously.
>
> > ... AbDiffuser's "folding" performance is bounded by IgFold. The model is ranked with another classifier trained with binding probability (line 316). It appears that the IgFold and classifier act as the model's "teacher." How does the model outperform its "teacher"? Why not try to use such a "teacher" directly, with MCMC, Bayesian optimization, etc?
>
> Indeed, the model’s folding capacity is upper bounded by IgFold folding performance when trained only on folded structures, though as we also show in the new experiments it’s also possible to train on crystal structures. Also, the classifier is not used as a teacher but strictly for evaluating how well the generative models model the binder distribution (note that the generative models are trained only on binders, while the classifier sees both binders and non-binders).
>
> Nevertheless, it is interesting to consider the spirit of your question. In general, there are two main arguments why employing a generative model to model the distribution is a better choice than using search:
>
> First, there is an intimate interaction between sequence and structure which motivates building a model that generates sequence and structure jointly: structural features are known to be quite informative of what amino acid types are acceptable in a given structural position (e.g., due to charge complementarity of nearby amino acid side chains). So the hypothesis is that generating structure as well, should make the model generate better sequences. This is confirmed by our experiments, since the structure+sequence models outperformed sequence-only models, even when they were trained on much more data.
>
> Second, a key pitfall of directly searching (MCMC/Bayesian opt) for Abs that abide to some properties according to some classifiers is that the classifier predictions p(y|x) are only trustworthy close to their training data distribution and do not reveal the all-important data likelihood p(x). Unfortunately, without controlling p(x) design is doomed to fail! That is because we can only trust what the classifier predicts if we have the ability to select likely samples according to the distribution it was trained on. The latter is what a good generative model (like a denoising diffusion) excels at, but is not achieved if we search over the space of Abs that the folding model and the classifier predict as positives.
>
> > During forward diffusion, is the residue type continuous or discrete? (...).
>
> The residue type is always discrete (both in forward and reverse processes) and is selected by sampling. In your example, if at t=500 the categorical probability distribution was (0.1, 0.1, 0.1, 0.1, 0.6), we would sample from it to produce the noised discrete sample which could result in a residue in any state (it would end up at state 1 with probability 0.1 and so on).
>
> > Are the priors computed over the dataset? What would be the performance if directly sampling from the sequence and structure prior?
>
> We compute the priors over the training set. But it could make sense to say compute the priors over all known antibody structures and sequences.
>
> Sampling the structure prior still gives you a random (Gaussian) point cloud. Atom positions in it are of course correlated, but as a whole it looks nothing like a true antibody. So the performance of any structural metrics is very poor.
>
> In sequences, the positions in the antibody framework regions can be highly conserved (e.g., we can have just a couple of different amino acid types observed in a particular AHo position). So randomly sampling the prior would give you a highly statistically similar sequence in the framework region (though it might not achieve high expression in vitro). CDR regions are much more diverse, especially CDR H3 and CDR L3. So there the sampling is more random. When we use the priors to sample sequence and structure for paired OAS, we get WD(Naturalness): 0.4004, WD(Closeness): 0.2969, WD(Stability): 0.3871, RMSD:  17.3944. These results are an order of magnitude worse than what we can achieve with AbDiffuser: WD(Naturalness): 0.0916 , WD(Closeness): 0.0520, WD(Stability): 0.0186, RMSD:  0.4962.
>
> > Random mutation approach success rate
>
> In short, this is not true as the dataset considered does not contain random mutations but combinations of carefully selected mutations determined by experts using a deep mutagenesis experiment (see Mason et al, DOI: 10.1038/s41551-021-00699-9). Due to the space limit of the rebuttal we will expand on this in the discussion period.

---

> > ### Author Response · Authors · 2023-08-10
> > **Random mutation success rate**
> >
> > > In line 314, a random mutation approach may generate 9k binders and 25k non-binders, indicating that random mutation causes a success rate of 9/(9+25)=26%. In line 66, AbDiffuser has a success rate of 22.2%. Although the number may increase to 57.1% by post processing, does this mean that "AbDiffuser performs worse than a random mutation baseline"?
> >
> > Details on why this is incorrect:
> >
> > 1. Random CDRH3 mutations taken from OAS are known to have binding rates close to 2.68% as reported by Shanehsazzadeh et al (DOI: 10.1101/2023.01.08.523187).
> >
> > 2. In contrast, the dataset considered does not contain random mutations but combinations of carefully selected mutations determined by experts using a deep mutagenesis experiment (see Mason et al, DOI: 10.1038/s41551-021-00699-9). The dataset was also built using noisy high throughput yeast display experiments and our analysis suggests that ~21% of the reported labels are wrong (deduced from sequences with both positive and negative labels, which we drop in pre-processing). So, if an unconditional generative model is trained on both binders and non-binders and exactly captures the training distribution, it is expected to achieve a binding rate of ~26% +/- 5%. (The SPR wet-lab experiments we performed are precise and are not susceptible to such issues.) One can use guidance or in silico filtering to increase the binding rate, but we cannot generally expect to do better without them.
> >
> > 3. Our analysis in Appendix K demonstrates that many of our binders validated in SPR experiments, including the best one, are the same number of mutations away from both the binder set and the non-binder set. This highlights that it is very easy to make wrong mutations, especially if the negative examples are not known (as is the case in our experiments).

---

> ### Author Response · Authors · 2023-08-19
>
> Dear Reviewer is6j, since the discussion period is coming to an end soon we wanted to ask if there are any remaining questions that we can address?

---

> > ### Comment · Reviewer_is6j · 2023-08-19
> >
> > I have read all the discussions and most of my concerns have been addressed. Thank you for your effort to make things clearer and I am willing to raise my score.

---

### Official Review · Reviewer_AFyF · 2023-07-06

**Soundness:** 4 excellent
**Presentation:** 4 excellent
**Contribution:** 3 good
**Rating:** 8
**Confidence:** 4

**Summary:**

Antibody design has an extreme importance for both fundamental and application biologic science. The submission intriduces an equivariant and physics-informed diffusion model called AbDiffuser

**Strengths:**

Originality: There are few tools that have already addressed the same problem (for example, DiffAb, https://github.com/luost26/diffab). However, due to the lack of systematic evaluation and comparison between the approaches, AbDiffuser remains useful and important new tool.
Quality: The submission provides both in silico and in vitro validation that is crucial for antibody design. Authors provide a detailed supplementary with clear explanation of used methods and approaches. However, the paper lack a proper discussion about potential weaknesses and limitations of AbDiffuser.
Clarity: The test is clear and all the sections are presented in structures manner.
Significance: The results are important and useful for biology and drug discovery in particular. AbDiffuser can be used by researchers for a fast and efficient design of antibodies to particular antigen.

**Weaknesses:**

Even though the paper provides comparison of AbDiffuser with other models, it lacks comparison to specific to antibody-antigen diffusion models (for example, https://github.com/luost26/diffab). It can be useful to discuss AbDiffuser compared to these tools. Also, there are several new protein-focused diffusion models such as RFdiffusion (https://github.com/RosettaCommons/RFdiffusion) or DiffDock-PP (https://github.com/ketatam/DiffDock-PP), the discussion of or comparison to can be valuable. I suggest extending the "Related Work" section to highlight the mentioned above issues.

**Questions:**

It can be useful to discuss AbDiffuser compared to other diffusion tools focused on proteins and antibodies such as RFdiffusion, DiffDock-PP, and DiffAb ((https://github.com/RosettaCommons/RFdiffusion, https://github.com/ketatam/DiffDock-PP, https://github.com/luost26/diffab)

**Limitations:**

Authors need to include a section about limitations and weaknesses of AbDiffuser as well as potential improvement of the method.

---

> ### Author Rebuttal · Authors · 2023-08-10
>
> Thank you for your review and recognising the potential of our work. We answer your comments below.
>
> > There are few tools that have already addressed the same problem (for example, DiffAb, https://github.com/luost26/diffab). (..) Even though the paper provides comparison of AbDiffuser with other models, it lacks comparison to specific to antibody-antigen diffusion models (for example, https://github.com/luost26/diffab). It can be useful to discuss AbDiffuser compared to these tools.
>
> From a method standpoint, DiffAb and similar tools (e.g., MEAN) aim to re-design the CDR loops in a known co-crystal structure, and not the full antibody as we do. Often, as in the case of DiffAb, they are only tested at re-generating one CDR loop at a time. We discuss the works using such approaches in more detail in the related work (Appendix A, line 650 and briefly in lines 41-42 of the main text).
>
> To test how well these methods work in comparison to AbDiffuser, we retrained DiffAb and dyMEAN for the HER2 binder design task (see our general response above). For completeness, we also added experiments on conditioned CDR redesign using the SAbDab split from DiffAb. The results showcase that in the conditional CDR redesign AbDiffuser performs much better in terms of CDR sequence recovery than DiffAb with an impressive 1.58x improvement in CDR H3 amino acid recovery.
>
> > However, the paper lacks a proper discussion about potential weaknesses and limitations of AbDiffuser.
>
> The two main limitations that we see are that 1) our method as is can only be applied to proteins that belong to a conserved fold (such as the Abs, TCRs, TIM-barrels, etc), and 2) that the original experiments focused on unconditional generation, so each new target will require a new dataset (obtained e.g., by a high throughput display experiment).  We mention these limitations in the paper, but we will make sure to highlight them further in the final version.
>
> We also note that the second limitation was mitigated by the new experiment we performed on SAbDab CDR inpainting task – the same experiment performed by DiffAb (see general response).
>
> > Also, there are several new protein-focused diffusion models such as RFdiffusion (https://github.com/RosettaCommons/RFdiffusion) or DiffDock-PP (https://github.com/ketatam/DiffDock-PP), the discussion of or comparison to can be valuable. I suggest extending the "Related Work" section to highlight the mentioned above issues.
>
> The paper discusses RFDiffusion and explains that the latter’s utility for Abs is limited as the method is not built to generate the sequence but only the protein backbone (which is less challenging for Abs). We will extend the discussion accordingly and include DiffDock-PP in it, explaining that there are interesting advances in protein design that address different problems from generation (that we focus on).

---

> > ### Comment · Reviewer_AFyF · 2023-08-17
> >
> > Thank you for the changes and additions to the paper. I am satisfied with the rebuttal and have no further comments to add.

---

### Author Rebuttal · Authors · 2023-08-10

We thank all reviewers for their thoughtful comments and suggestions. In response, we performed additional experiments, adding two state-of-the-art baselines to our current comparison and showcasing how AbDiffuser significantly outperforms previous methods in SAbDab CDR inpainting.

### HER2 binder generation task – DiffAb and dyMEAN (Table 3).

We found that DiffAb trained on HER2 binders focused on high-likelihood modes of the distribution and did not generate perfectly unique samples, but achieved good binding probabilities. To produce a head-to-head comparison, we also generated samples from AbDiffuser by removing the additional noise added during every step of the reverse process, similarly to what was proposed in RFDiffusion. As shown in the table below, the binding probability was greatly increased (surpassing DiffAb) at the cost of focusing more on high-likelihood modes.

| Model | \\(W_1(\text{Nat.})\downarrow\\) | \\(W_1(\text{Clo.})\downarrow\\) | \\(W_1(\text{Sta.})\downarrow\\) | \\(W_1(\text{PSH})\downarrow\\) | \\(W_1(\text{PPC})\downarrow\\) | \\(W_1(\text{PNC})\downarrow\\) |  \\(W_1(\text{CSP})\downarrow\\) | \\(W_1(\Delta G)\downarrow\\) | \\(\text{RMSD}\downarrow\\) | \\(p_{\text{bind}}\uparrow\\) | \\(\text{Uniq.}\uparrow \\) |
|---|---|---|---|---|---|---|---|---|---|---|---|
| DiffAb | 0.0074 | 0.0014 | 0.0498 | 0.5481 | 0.0097 | 0.0067 | 3.4647 | 6.7419  | 0.4151 | 0.8876 | 99.7% |
| AbDiffuser | 0.0013 | 0.0018 | 0.0028 | 0.4968 | 0.0205 | 0.0113 | 0.1588 | 6.4301 | 0.3822 | 0.5761 | 100% |
| AbDiffuser (side chains) | 0.0010 | 0.0005 | 0.0062 | 1.2909 | 0.0115 | 0.0029 | 0.0948 | 32.0464 | 0.4046 | 0.6848 | 100% |
| AbDiffuser (no noise) | 0.0008 | 0.0014 | 0.0265 | 2.5944 | 0.0206 | 0.0053 | 0.2378 | 15.2200 | 0.3345 | 0.9115 | 99.7% |
| AbDiffuser (side chains, no noise) | 0.0015 | 0.0024 | 0.0159 | 1.5043 | 0.0210 | 0.0126 | 0.5173 | 114.4841 | 0.6795 | 0.9436 | 91.4% |


We also trained dyMEAN on the HER2 dataset (code became available after the NeurIPS submission deadline). Unfortunately, we found that dyMEAN was very prone to collapse. Over 4 distinct runs, the model with the best validation loss always generated only a single example.

### Full-Ab generation on paired OAS - dyMEAN (Table 1).

In contrast to MEAN and DiffAb, the latest version of dyMEAN has been shown to be able to generate full antibodies. We thus also trained dyMEAN on the paired OAS generation (Table 1) and got the following results:

| Model | \\(W_1(\text{Nat.})\downarrow\\) | \\(W_1(\text{Clo.})\,\downarrow\\) | \\(W_1(\text{Sta.})\downarrow\\) | \\(W_1(\text{PSH})\downarrow\\) | \\(W_1(\text{PPC})\downarrow\\) | \\(W_1(\text{PNC})\downarrow\\) |  \\(W_1(\text{CSP})\downarrow\\) | \\(W_1(\Delta G)\downarrow\\)  | \\(\text{RMSD}\downarrow\\) | \\(\text{Uniq.}\uparrow \\) |
|---|---|---|---|---|---|---|---|---|---|---|
dyMEAN | 0.1319 | 0.1600 | 0.0423 | 3.9145 | 0.1566 | 0.2929 | 2.3711 | 601.1153 | 3.8157 | 58.2% |
AbDiffuser | 0.1979 | 0.0921 | 0.0662 | 2.3219 | 0.0314 | 0.0285 | 0.6662 | 13.3051 | 0.5230 | 100% |
AbDiffuser (side chains) | 0.0916 | 0.0520 | 0.0186 | 6.3166 | 0.0209 | 0.0754 | 0.8676 | 16.6117 | 0.4962 | 100% |

Only 58.2% of the 1k generated sequences were unique, which is problematic considering that paired OAS has 100k antibodies in the training set. Our results indicate that further advances would be needed to avoid mode collapse and to accurately model paired OAS with dyMEAN.

### SABDab CDR inpainting

We evaluated AbDiffuser on the SABDab co-crystal structure CDR inpainting task from DiffAb using their splits and experimental setup (Section 4.1 in https://www.biorxiv.org/content/10.1101/2022.07.10.499510v5.full.pdf). We conditioned AbDiffuser on the antigen by replacing the initial embedding MLP in APMixer by the output of a GNN layer that passes messages from the closest 10 antigen residues to the CDR residues. We will add further details in the supplement.

The table below shows that AbDiffuser outperformed Rosetta and DiffAb on amino-acid recovery with a wide margin. The CDR RMSD results are quite comparable between DiffAb and AbDiffuser, but importantly AbDiffuser does better than DiffAb on CDR H3 and CDR L3 which have the most variability in their structure and can influence binding the most. The good performance of AbDiffuser in amino-acid recovery could be attributed to the usefulness of AHo numbering and APMixer likely being better at modeling sequences than a GNN. Our ablation experiments on paired OAS generation (Table 1) corroborate this hypothesis.

| Model | AA CDR H1 | AA CDR H2  | AA CDR H3 | RMSD CDR H1 | RMSD CDR H2 |  RMSD CDR H3 | AA CDR L1 | AA CDR L2  | AA CDR L3 | RMSD CDR L1 | RMSD CDR L2 |  RMSD CDR L3 |
|---|---|---|---|---|---|---|---|---|---|---|---|---|
| Rosetta (RAbD) | 22.85% | 25.5% | 22.14% | 2.261 | 1.641 | 2.9 | 34.27% | 26.3% |  20.73% | 1.204 | 1.767 | 1.624 |
| DiffAb | 65.75% | 49.31% | 26.78% | 1.188 | 1.076 | 3.597 | 55.67% | 59.32% | 46.47% | 1.388 | 1.373 | 1.627 |
| AbDiffuser | 78.48% | 63.68% | 42.19%  | 1.507 | 1.296 | 3.431 | 83.84% | 90.35% | 70.98% | 1.268  | 1.433  | 1.312 |

AA stands for Amino Acid recovery and RMSD measures the CA position difference to the CDR CA positions in the original co-crystal structure.

DiffAb also included the estimated improvement in binding score (Rosetta ddG) in their metrics. Since Rosetta ddG has been proven to be an unreliable metric for determining binding affinity (see e.g., Figure 2b in https://pubs.acs.org/doi/10.1021/acs.jpcb.7b11367) we avoid using it for model comparison.

Note that we used the DiffAb split as it properly ensures that antibodies similar to the ones in the test set are removed from the training set (sequences are clustered for up to 50% CDR H3 identity). dyMEAN in their split does not ensure this non-overlap so their reported results are not comparable.

Additionally we attach a PDF with training and sampling pseudocode.

---

### Decision · Program_Chairs · 2023-09-21

**Decision:**

Accept (spotlight)

**Comment:**

This paper presents an antibody-specific diffusion model that can be used for antibody design, an important application in biological sciences. Reviewers appreciated the design of the method, which leverages antibody domain knowledge and physics-based constraints, and the extensive evaluation of the method, including experimental results validating the antibody designs.